# Mapping the breast cancer metastatic cascade onto ctDNA using genetic and epigenetic clonal tracking

George D. Cresswell [1,8], Daniel Nichol [1,8], Inmaculada Spiteri[1], Haider Tari [1,2], Luis Zapata [1], Timon Heide [1], Carlo C. Maley [3], Luca Magnani [4], Gaia Schiavon[5,7], Alan Ashworth[6], Peter Barry[5✉] & Andrea Sottoriva [1✉]

Circulating tumour DNA (ctDNA) allows tracking of the evolution of human cancers at high resolution, overcoming many limitations of tissue biopsies. However, exploiting ctDNA to determine how a patient's cancer is evolving in order to aid clinical decisions remains difficult. This is because ctDNA is a mix of fragmented alleles, and the contribution of different cancer deposits to ctDNA is largely unknown. Profiling ctDNA almost invariably requires prior knowledge of what genomic alterations to track. Here, we leverage on a rapid autopsy programme to demonstrate that unbiased genomic characterisation of several metastatic sites and concomitant ctDNA profiling at whole-genome resolution reveals the extent to which ctDNA is representative of widespread disease. We also present a methylation profiling method that allows tracking evolutionary changes in ctDNA at single-molecule resolution without prior knowledge. These results have critical implications for the use of liquid biopsies to monitor cancer evolution in humans and guide treatment.

[1] Evolutionary Genomics and Modelling Lab, Centre for Evolution and Cancer, The Institute of Cancer Research, London, UK. [2] Glioma Lab, The Institute of Cancer Research, London, UK. [3] Arizona Cancer Evolution Center, Biodesign Institute, Arizona State University, Tempe, AZ, USA. [4] Department of Surgery and Cancer, Imperial College London, London, UK. [5] Breast Unit, Royal Marsden Hospital, London, UK. [6] UCSF Helen Diller Family Comprehensive Cancer Center, 1450 3rd St, San Francisco, CA 94158, USA. [7] Present address: AstraZeneca, Oncology R&D, Cambridge, UK. [8] These authors contributed equally: George D. Cresswell, Daniel Nichol. ✉email: peter.barry@icr.ac.uk; andrea.sottoriva@icr.ac.uk

iquid biopsies, comprising profiling circulating tumour DNA (ctDNA) from the plasma of cancer patients, have changed the way in which we can study human malignancies[1]. Taking plasma samples instead of tissue biopsies is easier and less invasive, and several studies have demonstrated the clinical value of tracking somatic mutations in ctDNA (reviewed in ref. [1]). However, technical limitations and lack of quantitative, standardised measurements prevent ctDNA from being used systematically, accurately and robustly[2]. Specifically, we face the following problems:

(a) Plasma ctDNA is a mixture of alleles from distinct cancer cells and, unlike tissue biopsies, cannot be separated into single cells.

(b) Plasma ctDNA is often diluted by large quantities of cell-free DNA from normal cells, making liquid biopsies very impure.

(c) The extent to which ctDNA reflects different metastatic lesions in distinct regions of the body remains largely unknown.

These limitations imply that unbiased genomic profiling of ctDNA remains difficult and costly. This is why to date, profiling ctDNA invariably requires prior knowledge of recurrent and/or patient-specific alterations in human cancers. Studying the evolution of treatment resistance using ctDNA, for example, requires prior knowledge of the mutations and pathways involved, for example RAS/RAF pathway mutations for EGFR inhibitiors[3–6], or DNA repair back-mutations in prostate cancers treated with PARP inhibitors[7]. Also tracking tumour progression in ctDNA requires prior knowledge of what mutations were present in the primary tumour[8–10]. Moreover, measuring the contribution of different metastatic sites to ctDNA remains challenging because comprehensive profiling of metastatic lesions is possible only post-mortem.

In this study, we leveraged on a rapid autopsy programme called LEGACY (CCR3995, REC 13/LO/1535) as well as a clinical study on early breast cancer (REC 13/LO/1015) taking place at the Royal Marsden NHS Trust in London to perform an unbiased characterisation of ctDNA and concomitant comprehensive profiling of many detectable lesions from metastatic breast cancer patients. Using deep whole-genome sequencing, we revealed the extent to which ctDNA represents the different lesions in a given patient. We also developed a new methylation profiling approach applied to ctDNA that allows tracking the evolution of sub-populations of cancer cells in plasma at single-molecule resolution without prior knowledge of the mutational profiles of solid metastases.

## Results

**Parsimonious reconstruction of the metastatic cascade in breast cancer.** LEGACY (CCR3995) was a pilot programme with the aim of determining feasibility of patients donating tissues soon after death from metastatic breast cancer. It included the collection of a blood sample for both germline DNA analysis as well as plasma DNA for ctDNA extraction. REC approval (IRB) number was 13/LO/1535 (clinicaltrials.gov: NCT02126800). In this study we present the analysis of the metastatic cascade and ctDNA profile of LEGACY patients 1 and 2.

LEGACY patient 1: A 51-year-old previously healthy female had a 2-month history of right upper quadrant abdominal pain, weight loss and anorexia. Imaging investigations confirmed liver metastases and a likely right breast primary carcinoma with axillary and mediastinal nodal involvement, i.e. de novo Stage IV breast cancer. Asymptomatic pulmonary emboli were found. Right breast core biopsy revealed grade 2 invasive ductal

carcinoma (T3, P2, M1) ER 8/8, PR 6/8, HER2 negative. The clinical history was as follows (see Supplementary Fig. 1A for diagram):

- Month 1: First-line chemotherapy (epirubicin and cyclophosphamide)

  – partial response in liver, stable at other sites
  – then progressive disease

- Months 3–8: Second-line chemotherapy (weekly paclitaxel)

  – partial response especially in liver

- Months 9–14: Maintenance endocrine therapy (letrozole)

  – progressive disease with new peritoneal nodules and progression of liver metastases

- Months 15–16: Third-line chemotherapy (capecitabine), stopped due to toxicity

  – PET/CT stable disease (Month 17)

- Months 17–19: Maintenance endocrine therapy (exemestane)
- Month 19: Hypercalcaemic crisis
- Month 21: Died (hepatic failure primarily). Rapid donation performed within 4 h of death

We performed whole-genome sequencing at median 53.5× depth (45–145×) in 12 samples from the primary, ipsilateral axillary lymph nodes, liver, lung, diaphragm and ovary metastases, as well as matched normal blood DNA (61×, Fig. 1a). Copy number profiling revealed a striking copy neutral (diploid) genome-wide loss of heterozygosity (LOH), see Fig. 1b and Supplementary Fig. 2. This event was present in every analysed sample and was therefore truncal in the tumour. Copy neutral LOH results from losing one of the alleles and then copying the remaining allele (e.g. AB → AA). Interestingly, chromosome 20, X and 16p retained heterozygosity, indicating they were unaffected by the LOH event. Genome-wide copy neutral LOH events have also been recently reported to be recurrent in undifferentiated sarcomas[11]. Other copy number alterations reflected the typical changes previously seen in breast cancer, such as chromosome 1q, 8, 10p and 16p gains[12,13] (Supplementary Data 1). Somatic single-nucleotide variant (SNV) analysis revealed a *PIK3CA* E542K putative driver mutation that was clonal in all samples (Fig. 1c, Supplementary Data 2). This variant was validated in all samples using digital droplet PCR (Supplementary Data 3). We also identified a mutation in *ESR1* Y537N that was present only in the liver metastasis and likely conferred resistance to hormonal therapy. This is consistent with the clinical course of the patient, who progressed while on letrozole (initially) and then subsequently on exemestane, specifically in the liver. Consequently, the predominant site of progression and likely cause of death was liver failure. Interestingly we also report a second *ESR1* D538G mutation at 32% cancer cell fraction (CCF) in Ovary Met R2, indicating that strong selective pressures and consequent convergent evolution for *ESR1* mutants.

We then examined mutations and copy number profiles together. Notably, somatic mutations that happened before the truncal genome-wide LOH event are copied over to the other allele and hence are also found as homozygous in the tumour, whereas mutations that happened after are found in only one out of two alleles. Interestingly, all mutations in one out of two alleles (post-LOH) were either private to primary and regional (axillary) lymph nodes, or private to the metastases. This indicates that the copy neutral genome-wide LOH event was possibly what

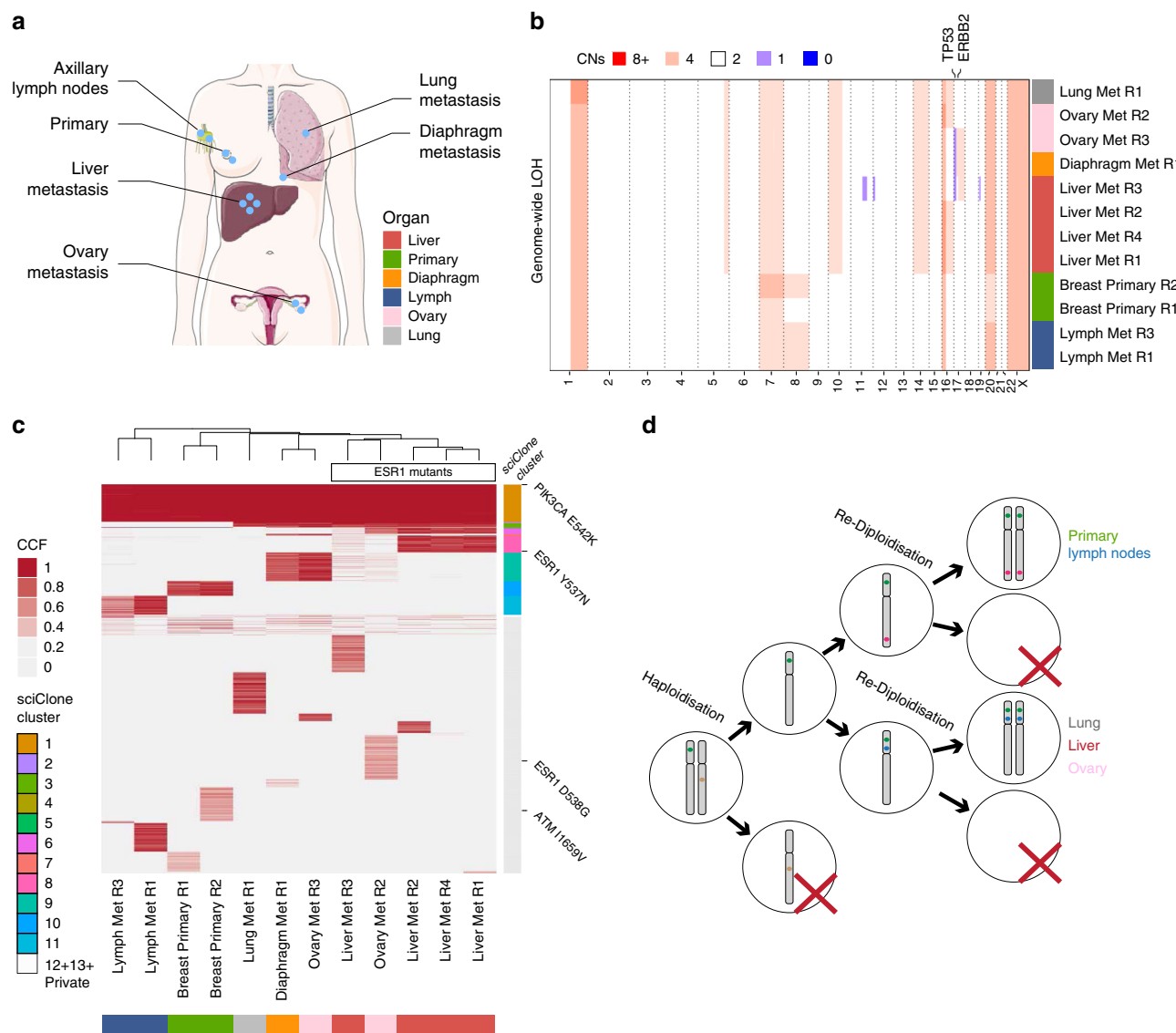

**Fig. 1 Genomic profiling analysis of LEGACY patient 1. a** Multiple samples from distinct metastatic deposits in different organs and the primary tumour were collected from this patient, together with blood germline reference (buffy coat) and plasma. **b** Copy number alterations analysis highlights genome-wide copy neutral LOH and overall homogeneous copy number profiles. Median total copy number in 1 Mb bins with a minimum mappability score of 0.8. **c** Single-nucleotide variant analysis identified a clonal PIK3CA driver mutation and convergent evolution for drug resistant ESR1 mutants. SNVs detected in more than one sample were clustered with sciClone (colour bar reported to the right of heatmap, see sciClone Cluster legend). **d** Diagram of inferred genome-wide copy neutral LOH event that can be explained by haploidisation followed by convergent re-diploidisation after a few cell divisions.

triggered the final malignant expansion. Strikingly, we also found a few somatic mutations in two copies (before re-diploidisation) that distinguished the two parallel lineages of primary and regional lymph nodes, from all the distant metastases (Supplementary Fig. 3). This complicated picture can be explained by a single haploidisation event where a diploid cell lost the maternal or paternal copy of most chromosomes. This probably led to high genomic instability and therefore high cell death. Stability was then restored in the haploid clone within a few cell divisions by two independent re-diploidisation events, one that gave rise to all the cells in the primary tumour and regional lymph nodes and one that gave rise to all the distant metastases (Fig. 1d). This suggests convergent evolution at the level of genome-wide copy neutral LOH, and to our knowledge is documented for the first time in this study.

Mutational signature analysis revealed that the predominant mutational process in all sites was Signature 1A which is the product of cytosine deamination at CpG sites due to ageing[14]. The only other detectable signature in the patient was Signature 2 (APOBEC) in the breast, lymph nodes and a single ovary sample (R2), indicating low levels of early APOBEC activity that may be diluted by an increasing mutational burden (Supplementary Fig. 4).

Previous studies on metastatic disease in prostate cancer[15] and breast cancer[16–18] reported extensive polyclonal seeding of metastases, as well as re-seeding from metastasis to primary, exposing a level of complexity in the metastatic cascade that, if true, would make this a clinically intractable problem. A more parsimonious analysis of the same data revealed that many of the complex patterns were often due to spurious low frequency variants or difficult to infer copy number states[19]. In our own reanalysis of ref. [16], we show that resolved copy number analysis and phylogenetic reconstruction identifies a much simpler metastatic cascade, where although there were polyclonal samples (samples containing more than one clone), seeding of metastatic

deposits was largely monoclonal in origin (Supplementary Fig. 5, Supplementary Materials and Methods). These results are consistent with a more recent analysis of primary-metastatic pairs in colorectal cancer[20]. In this regard, careful identification of genetically distinct subpopulations of cancer cells is crucial, and liberal subclonal classifications can instead produce over-complicated patterns[21] where the inferred evolutionary history of the tumour is driven by measurement noise. Moreover, the identification of the correct metastatic cascade is limited often by single samples taken from each metastatic site, instead of multi-region sampling from each metastatic deposit. To reconstruct the metastatic cascade in the LEGACY cohort we focused on mutations with a single copy number state in all sites (diploid), exploiting the hitchhiking principle to reconstruct the phylogenetic tumour tree. We then employed two separate clustering methods (sciClone and k-means), followed by manual curation to identify the correct subclonal clusters of mutations (see Methods).

Parsimonious reconstruction of the metastatic cascade in this patient revealed a relatively simple dissemination pattern where a single lineage from the primary tumour seeded all distant metastases very early during the evolution of the tumour (Fig. 2a, Supplementary Fig. 6, Supplementary Data 4). Lymph node dissemination also occurred early but independently. Early dissemination has been documented before both in breast[22] and colon cancer[20,23]. Hence, dissemination to the large majority of sites was consistent with monoclonal seeding (see Supplementary Fig. 7). However, 2/12 samples, one in the liver and one in the ovary were polyclonal in origin, suggesting exchange of seeding between these two metastatic sites. Specifically, a major subpopulation of cells in Liver Met R3 appears to have seeded a minor population of cells in Ovary Met R2 that is a closely related ancestor to the cell that seeded the entire Ovary Met R3 and the diaphragmatic metastasis (Fig. 2b, clones of interest highlighted, other clones in grey). Similarly, we identified a subpopulation of cells that form the majority of cells in Ovary Met R2 and that are closely related to the minor population in Liver Met R3, indicating that the minor clone in this sample was seeded by the major clone in Ovary Met R2. This lineage appears to have developed an ESR1 mutation and the clone is the only detectable population in the remaining liver samples, which were all monoclonally seeded (Liver Met R1, R2, and R4). The polyclonality of Liver Met R3 and Ovary Met R2 was supported by the presence of a 17p loss and 17q gain unique to the major clone of Liver Met R3 (Fig. 1b) and were detected in the minor clone of Ovary Met R2 in 20-24% of cancer cells. Additionally, the major clone of Ovary Met R2 contained an additional gain in 16p in 71% of cancer cells that was clonal in the monoclonal liver samples (see Supplementary Data 1). These CNAs represent independent support for the presence of these clones. We note that identification of sample polyclonality was not dependent on purity of the sample (see Supplementary Data 5).

These results indicate that the metastatic cascade in breast cancer is, although complex, a tractable evolutionary phenomenon that is driven mostly by monoclonal seeding, with rare polyclonal seeding events (see fully reconstructed metastatic cascade in Fig. 2c). In this individual there were two sites that showed evidence of being capable of harbouring genetically diverse subpopulations (Ovary Met R2 and Liver Met R3) and interestingly appear to have exchanged metastatic cells that have expanded and caused further spread. Focal 'niches' of polyclonality in specific metastatic locations have been documented previously in ovarian cancer[24]. Copy number profiles of the metastatic sites were largely homogeneous indicating that subclonal copy number alterations did not drive metastasis.

However, metastasis must have occurred almost immediately after the genome-wide loss-of-heterozygosity event (haploidisation and duplication) as we found no post-duplication mutations that were clonal across all samples. This suggests the haploidisation-duplication event may have been crucial to creating a metastatically competent lineage and may explain apparent homogeneity in copy number profiles of further metastatic sites.

To identify neoantigens present in each of the metastatic regions and the primary tumour we classified the HLA type for each region and then estimated the putative number of neoantigens (see Methods). We first observed that the HLA locus of patient 1 was identified as homozygous for most of the regions, confirming our previous observation of a haploidization event across the genome (Supplementary Fig. 2). Cases that were not homozygous for HLA were due to normal contamination in the sample. We identified only two neoantigens and those were present in all lesions, suggesting they were not able to elicit an immune response in the metastatic sites (Supplementary Data 6). We speculate that the general loss of heterozygosity in the primary tumour reduced the putative number of neoantigens and thus increased the chances of metastatic seeding. This loss of heterozygosity has been recently proposed as an immune evasion mechanism in ovarian[25], lung[26], and breast cancer[27].

LEGACY patient 2: A 35-year-old woman presented with a 4 cm left breast mass during the third trimester of her first pregnancy. She underwent left mastectomy and axillary lymphadenectomy for a 29 mm grade 3 and 23 mm grade 2 multifocal grade 2 ER positive, HER2 negative carcinoma with 10 of 14 nodes involved. Post-partum staging revealed de novo Stage IV with extensive bone and low volume lung metastases. The clinical history was (see Supplementary Fig. 1B for diagram):

- Months 3–7: First-line chemotherapy (capecitabine) plus denosumab

    – partial response on PET/CT; no new sites of disease

- Months 7–10: Maintenance endocrine therapy (goserelin plus tamoxifen)

    – rising tumour marker CA15.3. New liver disease on PET/CT

- Months 11–14: Second-line chemotherapy (epirubicin plus cyclophosphamide, 5 cycles)

    – mixed response at various sites but no new sites of disease on PET/CT after three cycles. Tumour markers fluctuating

- Month 14: Post-mastectomy chest wall radiotherapy

    – subtle progression in the liver on CT but stable disease at other soft tissue sites and bone disease on PET/CT. Tumour markers remain fluctuant

- Months 16–30: Third-line chemotherapy (paclitaxel)

    – tumour marker and PET/CT response until Month 30

- Months 35–43: Fourth-line targeted therapy with PARP inhibitor olaparib plus AKT inhibitor (phase I study)

    – partial response but progression by Month 41

- Month 47: Fifth-line chemotherapy (eribulin, 3 cycles)
- Months 48–49: Sixth-line chemotherapy (gemcitabine plus carboplatin, 2 cycles)

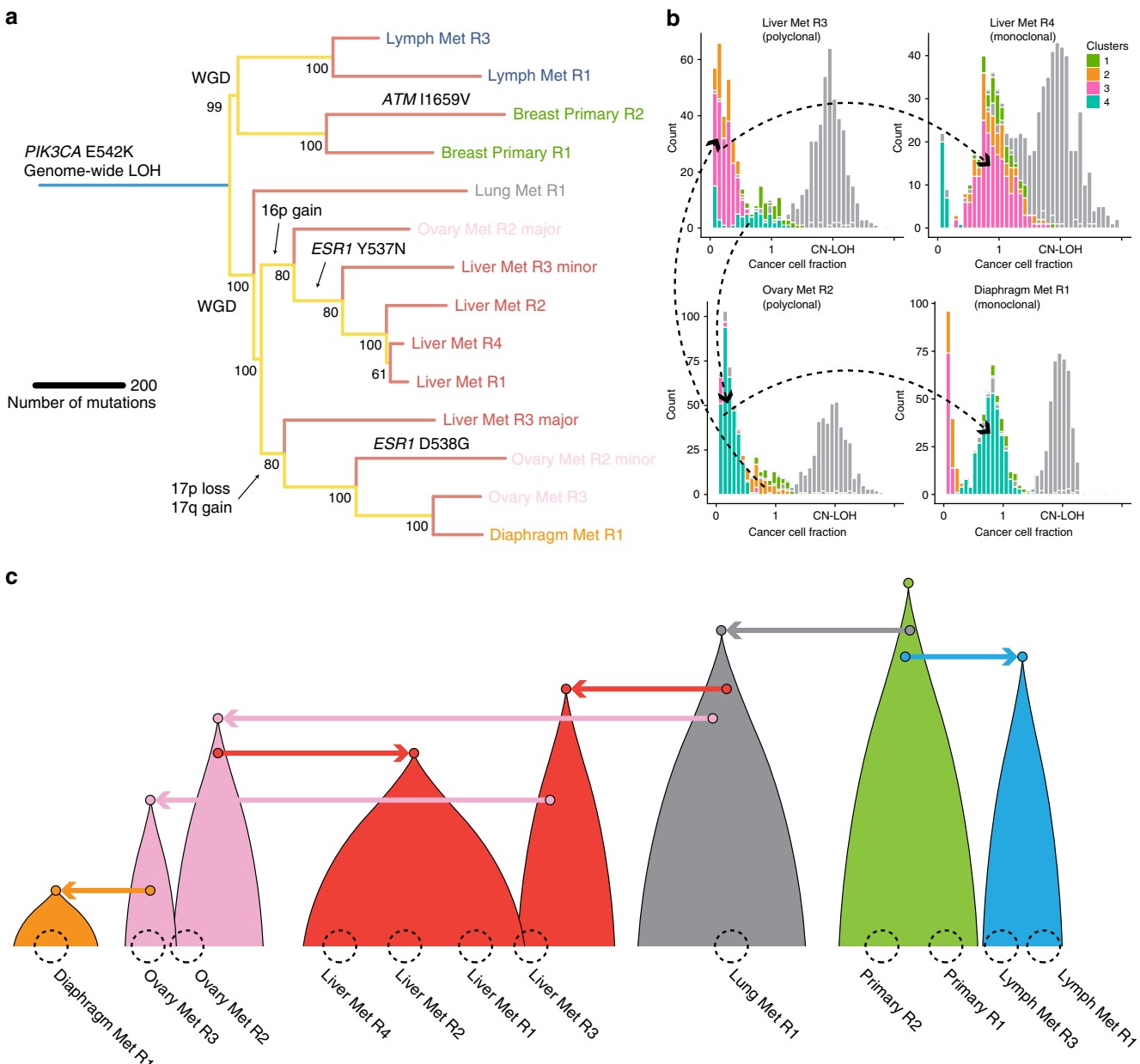

**Fig. 2 Clonal evolution analysis of LEGACY patient 1. a** Subclonal decomposition allows construction of the tumour clone tree from the sample tree, thus revealing the evolutionary history of the metastatic cascade. **b** Only 2/12 samples were found to be polyclonal (Liver Met R3 and Ovary Met R2) and indeed exchanged clones between each other. The same clones went on to seed distinct metastatic deposits in a monoclonal fashion. For example, the pink subclone in Liver Met R3 appears clonal in Liver Met R4. The turquoise subclone in Ovary Met R2 appears clonal in Diaphragm Met R1. The two clones remain subclonal but at distinct proportions in the two polyclonal samples Liver Met R3 and Ovary Met R2. Other clones are greyed for clarity. **c** Reconstructed metastatic cascade in breast cancer LEGACY patient 1 shows that the primary tumour (green) spread early to the lung (grey) and subsequently spread independently to draining lymph nodes (blue). From the lung, the cancer spread to Liver Met R3 (red) and Ovary Met R2 (pink) in two independent waves. Liver Met R3 then spread to Ovary Met R3 and subsequently to diaphragm (orange). Ovary Met R2 independently re-seeded back to the liver, giving rise to a large clone (Liver Met R1, R2, and R4). Hence, polyclonal seeding was evident in one liver and one ovary sample, but otherwise early monoclonal seeding was the dominant pattern of metastatic spread.

- Months 50–53: Seventh-line therapy with CDK4/6 inhibitor palbocliclib and exemestane (previously intolerant to fulvestrant)
- Month 53: Died (fulminant hepatic failure)

Whole-genome at median 95× depth (73–119×) was performed in nine samples from the para-aortic lymph nodes, lung and liver, in addition to normal DNA (blood) as previously (39×, Fig. 3a). Copy number alterations were largely homogeneous across metastatic sites and include common breast cancer copy number alterations such as chromosome 1q gain and 16q loss (Supplementary Data 1). A chromosome 8 gain was unique to Liver Met R3 (Fig. 3b). Mutational analysis reveals a clonal *PIK3CA* G1049R and a clonal *ERBB2* E770_A771insAYVM mutation (Fig. 3c, Supplementary Data 2). Chromosome 13 loss was identified in all samples. The heterozygous loss contains BRCA2 and RB1, the latter linked to resistance to CDK4/6 inhibitors[28]. Interestingly, the two samples with the highest number of private mutations were Liver Met R2 and R3, both of which had distinct truncating

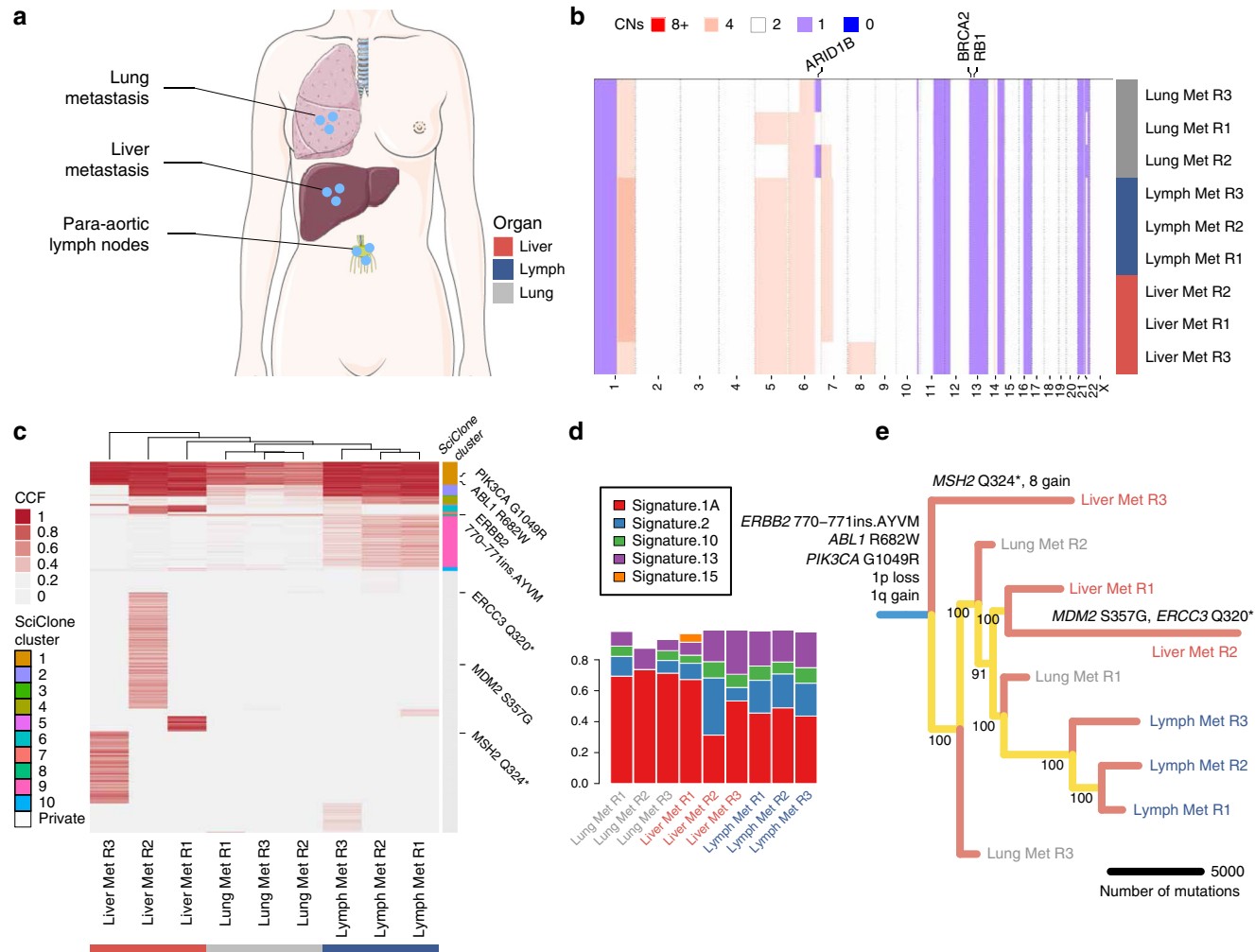

**Fig. 3 Genomic profiling and clonal evolution analysis of LEGACY patient 2. a** Multiple samples from distinct metastatic deposits in different organs were collected from this patient, together with blood germline reference (buffy coat) and plasma. **b** Copy number alterations analysis highlights largely homogeneous copy number profiles. **c** Single-nucleotide variants analysis identified clonal PIK3CA and ERBB2 putative driver mutations and multiple subclonal truncating mutations in DNA repair genes that seem to be responsible for the higher number of private mutations in Liver Met R2 and R3. **d** Mutational signature analysis highlights aging (signature 1A) but also APOBEC (signatures 2 and 13) as dominant mutational processes. **e** Clone tree shows early divergence for liver and lung metastases but largely monoclonal seeding of all lesions.

mutations in DNA repair genes (*ERCC3* and *MSH2*, respectively). Signatures 1A (age) and 13 (AID/APOBEC) were detected in all samples. Signature 2 (AID/APOBEC) was detected in all samples except for Lung Met R2 (Fig. 3d).

Phylogenetic reconstruction revealed that the para-aortic lymph node samples appear to have seeded from the lung metastases which may reflect the lymph drainage pattern from the lung. Liver Met R3 diverged early from the other metastatic samples and Liver Met R2 contained the highest burden of private mutations perhaps due to increased APOBEC expression and ERCC3 mutation. In this patient, according to the samples examined, each metastatic sample appeared monoclonal (Fig. 3e, Supplementary Fig. 8).

**ctDNA reflects the dynamics of metastasis and treatment resistance**. We performed whole-genome sequencing at 40× depth of the ctDNA in both patients. ctDNA from LEGACY patient 1 was particularly pure (79%) whereas patient 2 had a lower purity (33%). We performed somatic SNV genotyping using the mutations found in the tissue samples (therefore

including all mutations reported in Figs. 1c and 3c). We used these variants and multiple linear regression to determine the contribution of each tissue sample to ctDNA. We found that the plasma sample of Patient 1 predominantly reflected metastatic liver disease, especially the treatment-resistant ESR1 mutant clone, consistent with clinical course (Fig. 4a). This is further evidence that the ESR1 mutant clone was driving resistance by outcompeting the rest of the cancer cell population through proliferative advantage under hormone therapy. With more cell division and turnover, increased shedding of tumour DNA from the resistant clone into the bloodstream occurred. The proposed increase in cell proliferation of the resistant clone and its clinical progression as seen on PET/CT is consistent with liver failure as the primary cause of death. In total, the sum of contribution coefficients of the liver samples in Patient 1 was 0.86 and this was corroborated with the ESR1 mutation being present in an estimated 98% of cancer cells that shed DNA in the plasma. LEGACY patient 2 ctDNA reflected predominantly active metastatic sites in the lung and para-aortic lymph nodes (Fig. 4b). These patterns are also evident from the heatmap of SNVs when ctDNA was included (Supplementary Fig. 9).

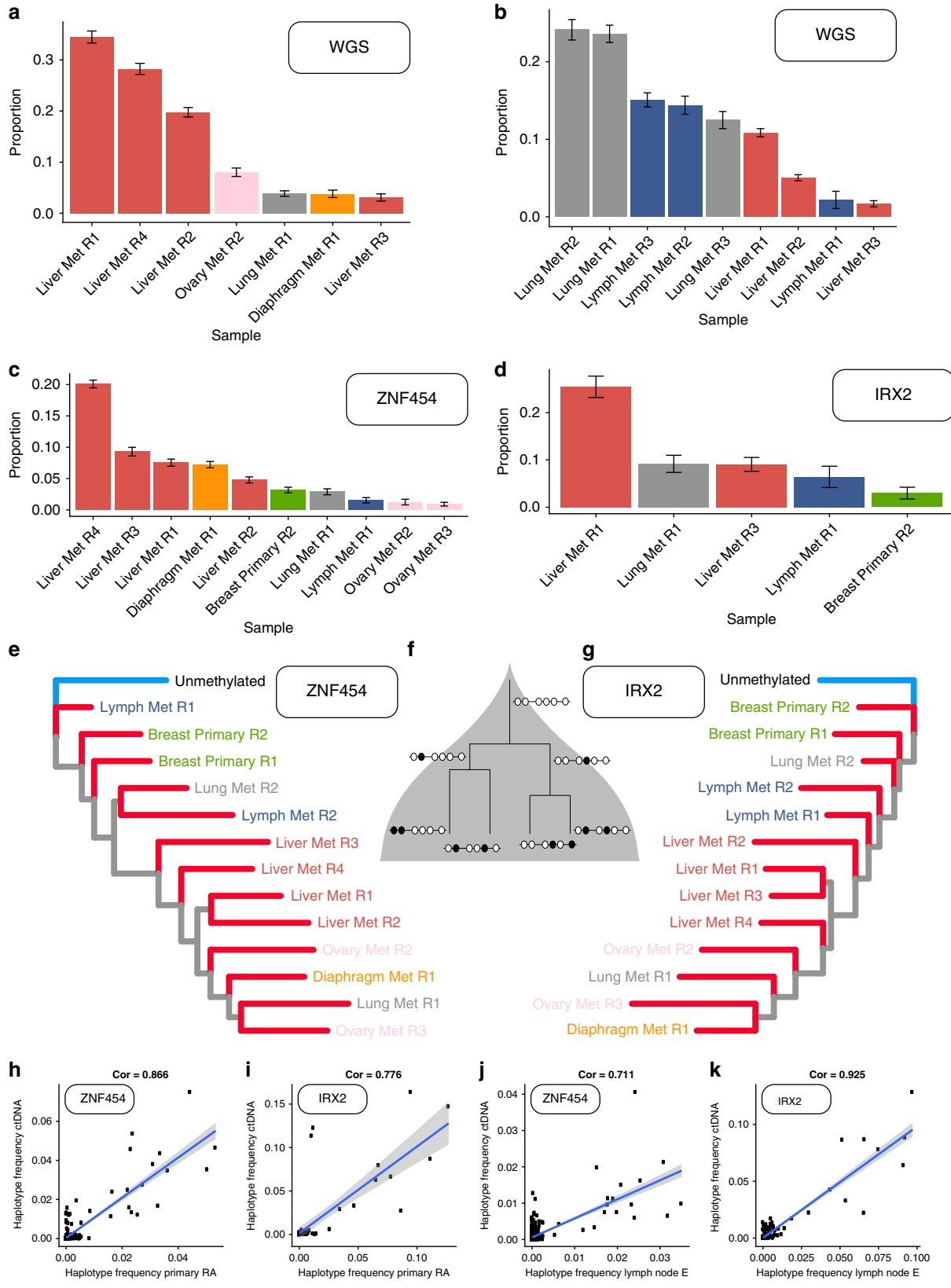

**Tracking clonal dynamics in ctDNA at single-molecule resolution using methylation clocks**. Due to the technical limitations of ctDNA samples, even high-depth whole-genome sequencing (60–100×) cannot capture the subclonal composition of cancer cell populations at high resolution, and only relatively large clones can be detected. Going deeper would require whole-genome profiling at >1000×, which is uneconomic even for research purposes. This is problematic because most studies that use prior knowledge to profile liquid biopsies report the identification of clinically relevant subclonal mutations in the plasma at very low frequency, often at variant allele frequency (VAF) < 10%[1]. Moreover, resolving the subclonal architecture of a single bulk

**Fig. 4 Tracking the metastatic cascade in the plasma of LEGACY patients.** Contribution of different tumour samples to ctDNA in LEGACY patient 1 (**a**) and patient 2 (**b**) using multiple linear regression analysis of somatic mutations. Similarly, contribution of tumour sites to ctDNA can be measured using methylation clock analysis (epimutations instead of nucleotide substitutions), as presented here for LEGACY patient 1 using two clock-like methylation regions that have been previously validated as subject to methylation drift, namely ZNF454 (**c**) and IRX2 (**d**). **e** With the same data we can also reconstruct the tumour phylogenetic tree using methylation clock haplotypes for ZNF454, corroborating the overall phylogenetic structure revealed by whole-genome sequencing. **f** This is because epimutations contain phylogenetic information about diverging cell lineages. **g** The same can be done for IRX2, leading to the construction of a consistent tree. Notably, methylation clock analysis can detect clones down to 1% prevalence and is >1000 less costly than whole-genome sequencing. **h–k** The same methylation clock analysis has been applied to an independent cohort of early breast cancer patients with matched primary, lymph node and ctDNA. High correlation (Pearson) between frequency of methylation haplotypes in the primary tumour of patient 6 was found for both clocks ZNF454 ($p = 4.94 \times 10^{-63}$) (**h**) and IRX2 ($p = 3.11 \times 10^{-13}$) (**i**). Also Patient 16 demonstrated high correlation between methylation patterns in ctDNA and a specific lymph nodal lesion, again for both ZNF454 ($p = 7.86 \times 10^{-60}$) (**j**) and IRX2 ($p = 7.51 \times 10^{-59}$) (**k**). *P* values refer to tests of no correlation.

---

sample such as plasma DNA remains challenging as it requires several correction steps and adjustments for copy number alterations and cellularity[29]. This is an unavoidable problem as ctDNA cannot be separated into smaller biopsies or single cells as solid tissue samples can. Spatial information of the distinct sub-clonal populations is therefore lost in the ctDNA, as the genome of cancer cells is broken into small pieces about a nucleosome interval long (~80–120 bp) and mixed together. The mutations that characterise a particular subclone invariably appear in different fragments from different genomic regions, and patching together these fragments from the same clone to reconstruct single haplotypes is de facto impossible.

To overcome this limitation, we developed a novel single-molecule methylation profiling assay applicable to ctDNA. Single CpG methylation changes (epimutations) are due to methyltransferase errors at cell division and occur 1000 to 10,000 times more frequently than DNA point mutations[30]. This implies that multiple somatic epimutations can occur close together in a so-called CpG island, to the extent that more than one CpG change can be found in the same piece of fragmented ctDNA, an event that almost never happens for point mutations and that allows clone haplotyping. Profiling somatic epimutations that are in the same DNA fragment can be done with standard Illumina sequencing (e.g. 150 bp pair-ended), thus permitting haplotype reconstruction using CpG methylation changes from standard bisulfite sequencing. Each haplotype from a specific CpG island in the genome is a 'barcode' sequence of zeros (non-methylated) and ones (methylated) that corresponds to an identifiable cancer cell lineage in the plasma. Several studies by us and others demonstrated that, by leveraging on epigenetic drift[31,32], passenger methylation haplotypes can be used to track the ancestry of normal somatic cells, as well as to profile the genealogy of cancer cells from solid malignancies such as colorectal cancer[23,26,27]. The idea is to use methylation as a 'molecular clock' to trace ancestry of somatic cell populations over time at single-molecule resolution. This concept, applied to solid tissues, has been around for a long time and has been pioneered by Shibata, Tavaré and colleagues, who used first microsatellite repeats[33] and then methylation patterns[31,32,34–36] as clocks in both normal and cancer tissues. The use of somatic mutations as molecular clocks has also been reported more recently with next-generation sequencing using mutational signatures[14]. Importantly, however, the resolution of methylation clocks is several orders of magnitude higher than point mutation clocks, such as mutational signature 1 (ageing), as the mutation rate is orders of magnitude higher[36]. In this study we extend methylation clocks to work on ctDNA in the plasma of cancer patients.

We previously identified two CpG island locations in the genome (ZNF454 and IRX2) that are amenable to be used as methylation clocks in cancer[30] (see Material and methods). Methylation data in healthy tissues from autopsy samples of

individuals with different ages for these loci have been analysed previously[30]. Specifically, methylation in two CpG islands in the IRX2 and ZNF454 loci increases linearly with age in dividing tissues, but remains stably low in non-dividing tissues. In this study we extended the assay and designed a new protocol that allows us to profile a shorter version of these clock regions in order to fit them within an average size plasma DNA fragment of ~80–120 bp in length (see Methods). We performed high-depth targeted bisulfite sequencing of these two methylation clocks (median 16,000× for ZNF454 and 22,000× for IRX2) in a total of 13 tissue samples and the ctDNA of LEGACY patient 1. We identified a large plethora of methylation haplotypes that did not show signs of hypermethylation (see Methods, Supplementary Fig. 10, Supplementary Data 7) and could be therefore used to trace ancestry. Moreover, thanks to the possibility of haplotyping epimutations, the contamination from non-cancer cells (e.g. peripheral blood lymphocytes) in cell-free DNA of cancer patients can be filtered out as blood cells have low methylation levels across the whole CpG island of clocks[30], and hence unmethylated haplotypes were excluded from our analysis.

Methylation levels from the tissue were conserved in the ctDNA, indicating that methylation clocks are a stable marker of cell fate. We applied the same regression method that we used to calculate the contribution of different sites to ctDNA from whole-genome sequencing data. We found that the epimutations largely recapitulated what we reported using nucleotide mutations from whole-genome sequencing, confirming that metastatic disease in the liver dominated plasma DNA (Fig. 4c, d). This demonstrates the power of methylation clocks in tracing ancestral cell populations applied for the first time to ctDNA. It is also important to note that these high-resolution single-molecule data cost over 1000 less to generate than the respective whole-genome sequencing, and moreover allow us to detect clonal populations identified by methylation haplotype families with prevalence as low or even lower than 1%. Unfortunately, methylation clock data from LEGACY patient 2 could not be used as they showed signs of hypermethylation in all sites (see Supplementary Fig. 11).

To further compare methylation clocks with whole-genome data, we also reconstructed a sample tree of the solid biopsies using methylation haplotypes (to be conservative in our phylogenetic analysis we excluded haplotypes with prevalence <1%). We confirmed that the ancestry recovered with epimutations in ZNF454 was highly concordant with the one recovered with nucleotide mutations from whole-genome sequencing (Fig. 4e). This is because epimutations naturally encode the genealogy of divergent lineages (cartoon in Fig. 4f). A very similar tree structure can be recovered using IRX2, demonstrating the robustness of methylation clock profiling in terms of phylogenetic reconstruction (Fig. 4g). Hence, three completely independent genomic profiling assays all inferred similar genealogic structures of the metastatic cascade.

To validate these findings, we profiled epimutations in the same methylation clocks of tissues and ctDNA from an orthogonal cohort of breast cancer patients with lymph node spread but no sign of distant metastases which we previously characterised with whole-exome and targeted sequencing[22]. Of those patients we profiled multiple samples from the primary and multiple lymph nodes. As those patients did not show sign of distant metastatic deposits, only a few samples per patient were available and hence linear regression as performed for LEGACY patient 1 would not be informative. Instead, we compared the frequency of methylation haplotypes in different solid tissue samples vs ctDNA to verify their concordance (e.g. proportion of reads with haplotype 001010011 in the solid tissue sample vs ctDNA). We calculated the correlation of methylation haplotype frequencies (VAF of a given binary string methylation haplotype) in each pair of tissue sample vs matched ctDNA and found very high correlations, with the two clocks again being concordant (Fig. 4h, i for Patient 6 and 4j, k for Patient 16, see Supplementary Fig. 12 for all correlations). This shows that the frequency of a methylation clone (haplotype) in the tumour tissue could be inferred from ctDNA with the presented methylation assay. This again confirms the power of methylation clock profiling for tracking clonal evolution in the plasma of cancer patients.

## Discussion

Understanding the time-course dynamics of the metastatic cascade is critical to cancer progression. Recent studies have highlighted overwhelming complexity of this cascade, posing a serious problem of how to translate those findings in the clinic. However, we argue that a considerable part of these findings is driven by the use of lower-resolution data, noise, and by inevitable limitations of current bioinformatics methods, which do not account for all the confounding factors of tumour evolution in space and time.

We show that metastatic progression is instead often a tractable problem with the right data and models. Unravelling the metastatic cascade can be interesting biologically, but how can we exploit this knowledge prospectively in the clinic?

We report the genetic evolution of two HR+/HER2− breast cancer patients who presented with de novo metastatic disease. In the first one, a clonal PIK3CA driver mutation and convergent evolution for drug-resistant ESR1 mutants evolved under the selective therapeutic pressure of endocrine therapy. Interestingly, we were also able to profile multiple samples from the primary tumour of this patient, taken at the same time as the other metastatic samples. This is rare in the context of autopsies as normally the primary is resected earlier and, even when available, cannot be compared within the same time point with the rest of the metastatic cascade. In this case the primary was not resected as deemed not clinically active, providing an opportunity for concomitant sampling of primary and metastatic deposits. Reconstruction of the metastatic cascade revealed that early monoclonal seeding was the dominant pattern of metastatic spread, as previously reported in colorectal cancer[20,23], with evidence of polyclonal seeding restricted to one liver and one ovary sample. In the second patient, treated with standard systemic agents as well as targeted therapy, monoclonal seeding was also dominant, with a pattern of early metastatic seeding to lung and liver and a single (monophyletic) metastatic clone invading the para-aortic lymph nodes. Of particular interest were subclonal mutations in DNA repair genes and concomitant increase in the number of mutations in the respective tree branches. We acknowledge that the limited incidence of polyclonality in the samples examined may not imply that this phenomenon is unimportant clinically, as this did contribute to the spreading of subpopulations in clinically active metastatic sites. Assessing

polyclonality vs monoclonality of metastatic seeding is crucial for a correct understanding of the metastatic cascade, especially when accumulating evidence from patient samples and the model system paints a complex and often discordant picture of a mixture of both monoclonal and polyclonal seeding[20,37,38].

Utilising ctDNA to track metastatic progression holds immense promise, but genomic profiling assays that focus on clonal evolution are missing. Here we propose a novel methylation assay that enables the tracking of the genealogy of cancer cell populations in the plasma of cancer patients at single-molecule resolution without prior knowledge using epimutations instead of nucleotide substitutions. Strikingly, the inference performed with whole-genome sequencing from the plasma to identify metastatic deposits contributing to plasma DNA was highly concordant with the one performed with methylation clocks. This methodology allows tracking cancer clones in the plasma at unprecedented resolution and 100–1000 times cheaper than standard genomic profiling with whole-exome or whole-genome sequencing. Additional studies will be needed to further develop this technique and extend this work to other tissue types and additional molecular clock loci. However, we envisage this technology could be used for longitudinal tracking of cancers under therapy.

## Methods

**Post-mortem sample collection**. All samples were retrieved from patients within 4 h of death. Large (6–8 mm) punch biopsies of solid organ metastases or whole lymph nodes were rapidly sampled, split and snap frozen in liquid nitrogen and stored at −80 °C prior to nucleic acid extraction and sequencing; adjacent samples from the same sites were preserved in 10% phosphate-buffered formaldehyde and processed for paraffin embedding and routine haematoxylin and eosin staining and immunohistochemistry. In preparation for nucleic acid extraction frozen section analysis of slices flanking the samples were also stained to assess the proportion of tumour for the entire extract.

**Sample preparation and sequencing**. Ten 20 μm cryo-sections from each sample were used for DNA extraction using the DNeasy kit (Qiagen) following the manufacturer's instructions. DNA samples were quantified using the Qubit dsDNA HS assay (Invitrogen) and 500 ng were sent to GATC Biotech Ltd (Germany) for library preparation and whole-genome sequencing. Libraries were sequenced on a HiSeq2500 (Illumina) to a median coverage of 60×.

**Methylation molecular clocks**. Two different loci on chromosome 5 were used as methylation molecular clocks: IRX2 (a 201 bp locus containing 9 CpGs) and ZNF454 (a 200bp locus bearing 16 CpGs). These loci were shown to behave as clocks in a previous publication[30], as shown by linear correlation with age in dividing tissues in Supplementary Figs. S2 and S5 of ref. [30], respectively. We also profiled a shorter version of ZNF454, containing 14 CpGs but which is amenable to ctDNA analysis where DNA fragments are particularly small. For each of the above samples, 50 ng of genomic DNA was bisulfite-converted using the EZ Methylation Direct kit (Zymo) following the manufacturer's recommendations. The resulting ssDNA was quantified with the Qubit ssDNA assay (Invitrogen). The molecular clocks were PCR amplified using the following primers: IRX2-fwd: GTATATTT TGTTAGGATTGGAGT, IRX2-rev: CATAAAACCCACATCTCTTTCAA, ZNF454-fwd: GGGAGGTTAGTTTAGGGGAG, ZNF454-rev: TTCTATTAC CTTCCAAACCTTTT. Between 1 and 5 ng of ssDNA was used to amplify each molecular clock in 25 μl PCR reactions made up from 12.5 μl of Kapa2G Robust ready mix (Kapa), 2 μl of the forward and reverse primer mix (10 μM), 1 μl of MgCl$_2$ (50 mM), 1.5 μl of DNA and 8 μl of DNase/RNase free water. The thermal profile of the PCR reaction was as follows: 95 °C/2 min, [95 °C/10 s, 58 °C/10 s, 72 °C/35 s] × 30, 72 °C/2 min. PCR products were purified with the QIAquick kit (Qiagen) and quantified with the Qubit dsDNA HS assay (Invitrogen). For each sample, equal amounts from both molecular clock products were pooled and Illumina-compatible libraries were prepared using the NEBnext Ultra II kit (NEB) as follows: 25 μl (approximately 10–30 ng) from each pool was mixed with 1.5 μl of the NEBnext Ultra II End Prep Enzyme Mix and 3.5 μl of the NEBnext Ultra II End Prep Reaction buffer. After 30 min incubation at 20 °C followed by 30 min incubation at 65 °C, 1.25 μl of the NEBnext Ultra II Adaptor (diluted five-fold) was added to the reactions. The NEBnext Ultra II Ligation Master Mix (15 μl) and NEBnext Ligation Enhancer (1 μl) were added to the previous reaction mixtures and incubated at 20 °C for 15 min. Further, 1.5 μl of USER enzyme was added to the ligation mixtures and then incubated at 37 °C for 15 min. The reactions were then purified with 1.4× SPRI-beads prior to PCR amplification. PCR reactions were prepared with 10.5 μl of ligated DNA fragments, 12.5 μl of NEBnext Ultra II Q5 Master Mix, 1 μl of i7-NEBnext Dual-Index Primer and 1 μl of i5-NEBnext Dual-Index Primer. The

PCR cycling conditions were as follows: 98 °C/30 s, [98 °C/10 s, 65 °C/75 s] × 6, 65 °C/5 min. Libraries were purified with 1.2× SPRI-beads and quality control was done on DNA-HS Bioanalyser Chips (Agilent). Finally, equimolar amounts of each library were mixed and the resulting pool was sequenced using MiSeq 300-cycle-v2 reagents (Illumina).

**Bioinformatics analysis: whole-genome sequencing.** FASTQ files were trimmed for adaptor content using skewer[39] with a minimum length allowed after trimming of 35 bp, only on reads with a minimum mean quality of 10 and with the filter to remove highly degenerative reads (-l 35 -Q 10 -n). Trimmed reads were aligned to hg38 (GRCh38) using bwa mem[40]. Sam files were sorted, compressed to bam files and duplicates were marked using Picard tools (http://broadinstitute.github.io/picard/). Bam files were indexed using samtools[41].

Mutations in each sample were called firstly using Mutect2 (ref. [42]). Variants were only kept if the coverage in both the tumour and normal was greater than 10 reads and the variant was present in three or more reads in the tumour. The variant must have the genotype '0/0' in the normal but not in the tumour. Mutations with the flag 'artifact_in_normal' were kept but variants called in each tumour sample were removed if their VAF was less than ten times greater than in the normal blood sample.

Per sample results for each patient were then merged and used as input for Platypus[43] that was run in genotyping mode. The following filtering criteria were used to filter variants after platypus genotyping: (i) only variants with Platypus filter PASS, alleleBias, Q20, QD, SC and HapScore were kept; (ii) minimum coverage and genotype quality of 10 was required for all samples; (iii) minimum of 3 reads covering the variant in at least one of the tumour samples per patient was required; (iv) the highest VAF in the tumour samples must be 10 times greater than the VAF in the normal; and (v) the germline sample must have a genotype of '0/0' and at least one tumour sample must not have a genotype of '0/0'. Variants were annotated using VEP[44].

Sequenza was used to identify heterozygous germline single-nucleotide polymorphisms (SNPs) in the bam files (0.4–0.6 allele frequency in the matched normal sample) and normalise depth ratios for GC content[45]. Loci were filtered for a minimum of 25 reads in the matched normal sample. Log2 ratios (LRR) were derived from the depth ratios by calculating the log (base 2) of the depth ratio and subtracting by the global median. LRR outliers were smoothed using CGHcall[46]. The mirrored allele frequencies of the heterozygous loci for each chromosome arm were segmented using piece-wise constant fitting (PCF)[47]. If B-allele frequency (BAF) values in segments are considered not to be drawn from a normal distribution expected in allele balance (BAF = 0.5, Kolmogorov–Smirnov test, $p <$ 0.05), a two-component Gaussian mixture model was fitted to the BAF values of the segment utilising mixtools version 1.0.4 (ref. [48]), in order to estimate BAFs representative of the major allele. The fixed standard deviation used for the Kolmogorov–Smirnov test is estimated from an initial pass of two-component Gaussian mixture modelling on all segments and also is used to restrict a grid search of parameters for modelling allelic imbalance (code available as an R package at www.github.com/georgecresswell/MiMMAl). Major allele BAF and the LRR of the genome segments were used as input for ASCAT[49] to estimate tumour purity and ploidy. The purity and ploidy of the ASCAT solution was used to assess the clonality of each segment using the Battenberg methodology[50]. If a segment was considered subclonal, the copy number state with the highest prevalence was taken.

For each variant the local total copy number state, VAF and sample purity was used to estimate CCF[51]. Purity estimated from copy number analysis was used. The number of alleles mutated was assumed to be 1 to avoid overcalling subclonality.

**Mutational signature analysis.** SNVs with a CCF of 0.2 or greater were used to calculate mutational signatures using deconstructSigs[52] using the signatures from Alexandrov et al.[53] as a reference and a minimum signature contribution of 0.05. On average, the number of mutations used for signature analysis in Patient 1 was 2429 (range 2073–3023) and in Patient 2 was 8884 (range 4484–17268).

**Phylogenetic analysis.** Binarised mutations were then used to create sample trees (CCF ≥ 0.2). Sample trees were constructed using maximum parsimony using phangorn[54] and the parsimony ratchet method. The tree is then rooted and edge length is determined by the ACCTRAN criterion.

**Subclonal reconstruction.** To ensure mutations were reliable prior to inferring potential subclones we filtered them for clonal diploid regions (homozygous diploid in Patient 1). In total, approximately 65% of the genome of Patient 1 and 59% of the genome of Patient 2 were clonally diploid (when considering 1 Mb bins with mappability greater than 0.8, see Figs. 1b and 3b). We used only diploid genomic regions for subclonal analysis, leveraging on the hitchhiking principle applied to cancer genomes (since cancers are made of somatic, asexually dividing cells). Additionally, for each mutation we calculated the average mappability of the region ±25 bp of each mutation from a bigWig generated for hg38 using the gem-mappability tool in the GEM library[55] and bigWigAverageOverBed[56]. We required that the average mappability value in these bins to be 1 (uniquely mappable). We also removed mutations that overlapped with simple repeats and

low complexity regions in accordance with annotations from RepeatMasker (http://www.repeatmasker.org).

These strictly filtered mutations were used as input for sciClone[57]. Only mutations with a CCF ≥ 0.2 in two or more samples were used as we were primarily interested in mutations subclonal in one sample and clonal/subclonal in another as these would be informative of the metastatic cascade, as opposed to private subclones. sciClone was run with a maximum number of clusters determined as twice the number of tumour samples, minus one. This is the number of edges in a tree of double the number of samples taken without private edges (i.e. sciClone will be able to detect all shared clusters that would be present in a clone tree with twice as many clones as tumour samples).

After manual inspection $k$-means clustering was also performed on Patient 1 ($k = 12$) and revealed that sciClone cluster 9 could be split into two clusters as a subset of the mutations had a high CCF in Liver Met R3 ($k$-means cluster 8) and the remainder of the cluster 9 mutations were largely absent in this sample. Using both independent clustering analyses we determined that Ovary Met R2 and Liver Met R3 were polyclonal. Mutations unique to the major clone in Liver Met R3 were defined as belonging to $k$-means cluster 8, and mutations unique to the minor clone in the sample were defined as $k$-means cluster 7 (containing all mutations in sciClone clusters 6 and 8). Mutations in the major clone in Ovary Met R2 were defined by sciClone cluster 6 (as $k$-means clustering did not re-identify this cluster) and the minor clone mutations were defined as belonging to $k$-means clusters 8 and 9 (containing all mutations in sciClone clusters 7 and 9, bar a single mutation in sciClone cluster 9). These definitions were used to separate the mutations in these polyclonal samples. Private mutations were also assigned in Ovary Met R2 and Liver Met R3 according to the distance of their CCF to the mean CCF of the clones in these samples, or were determined to be in both clones if their distance was closer to CCF = 1 than the mean of the major clone. These samples 'split' into major and minor subclones were used to generate a phylogenetic tree using the same method as described above.

**Neoantigen calling.** Polysolver V1.0 was run on each of the BAM files coming from the different sites of Patient 1 to predict the HLA type. Then, Pvactools v1.4.4 was run using the previously classified HLA types to identify the putative neoantigens present in each region. Both tools were run using default parameters.

**Methylation clocks analysis.** FASTQ files from the methylation clock assays were trimmed as described above. Paired reads were then aligned to hg19 using Bismark[58]. Bismark methylation extractor was then used to extract methylation states of possible CpG sites from the original top and bottom strands. For each molecular clock loci called CpGs are identified. CpGs are used for analysis if they have a total number of calls for a position (methylated plus unmethylated) greater than or equal to 1000. Reads with a call missing for a genomic position that passes this coverage filter are removed to leave only complete reads with a CpG call on all locations.

In each tumour sample reads are required to have at least one methylated CpG site and reads can only have a maximum of 80% methylation to remove reads that are likely produced by cells that have a low turnover (normal cells) and clocks that have reached saturation and are therefore non-informative, respectively. Next, for phylogenetic analysis specifically, remaining haplotypes (reads) with an overall abundance of 1% or less are removed due to their rarity. For each tumour sample 100 random haplotypes are selected with replacement and an additional set of 100 'synthetic' unmethylated haplotypes are created as a reference for each case. A similarity measurement is then performed pairwise between tumour samples and the unmethylated reference as used previously[30]. In brief, the Hamming distance of each haplotype combination between the two samples is measured and the shortest distance of all these combinations is recorded and the haplotype pair is removed from consideration. This is performed iteratively until all haplotype pairs have been removed and the Hamming distances of the chosen pairs are summed. This similarity measurement between all tumour samples and the unmethylated reference is used to create a Neighbour Joining tree using phangorn[54].

Haplotype frequency comparisons were performed following filtering for hypo- and hypermethylation and tests of no correlation were performed using cor.test in R.

**Contribution of sites to ctDNA.** ctDNA WGS bam files were processed identically to the tissue samples to generate bam files and copy number profiles. Mutations called in the tissues samples were then searched for in the ctDNA bam file using platypus in genotyping mode. Copy number analysis was used to determine local copy number and purity for calculating CCF as described above and CCFs were used to perform a linear regression to determine organ contributions to the ctDNA. Unfortunately in Patient 2 the callSubclones function in Battenberg failed; therefore, the purity was estimated from the ASCAT fit generated by the fit.copy.number function and we subset only clonal diploid regions in the tissue samples to assess organ contributions. Frequencies of haplotypes were calculated after filtering for a minimum of a one methylated site and less than 80% methylation and used for linear regression.

Model selection for the linear regression was performed by carrying out a non-negative least-squares estimation using the nnls R package. We begin by carrying a

forward stepwise regression introducing a parameter in the model that maximises adjusted $R^2$. We then recursively remove non-significant ($p > 0.05$) variables by removing the least significant variable.

**Reanalysis of Hoadley et al. (2016).** In their 2016 paper Hoadley et al.[16] explored the dynamics of metastatic dissemination in two patients with triple-negative metastatic breast cancer recruited as part of a rapid autopsy scheme. From the first of these patients, five tumour samples (primary, spinal, adrenal gland, liver and lung) were obtained and from the second six tumour samples (primary, rib, kidney, brain, liver and lung) were obtained. A subclonal deconvolution was performed for both patients and clone trees constructed using the tool ClonEvol[59]. We reanalysed the mutations identified in the original paper using sciClone[57] and with more careful and stricter criteria.

The pairwise VAF distributions among the samples from each patient are shown in Supplementary Fig. 5A and B, respectively. Here the mutations are coloured according to the cluster assigned to them by sciClone. Both the mutational frequencies and the clusterings are as reported by Hoadley et. al. In Supplementary Fig. 5A, by projecting the clusters onto the sample tree, we see that the clustering is entirely compatible with the sample tree phylogeny, although the mutations private to the liver sample are split into two separate clusters. Such a split can occur where a subclone is under selection and expanding within the population. However, where mutations exist in regions of copy gain, they can appear to be present at a higher frequency that expect under neutral growth. Under such circumstances, subclonal deconvolution algorithms can split clusters, potentially indicating an expanding subclone where none exists. Again, we must critically evaluate the evidence provided by a subclonal deconvolution to determine if the clone is spurious or not. Here, we examined the depth ratios (not shown) used to determine the copy number at the sites of the mutations in cluster 9. We found no evidence to suggest that the copy number differs from that for the mutations in other clusters and conclude the existence of an expanding subclone.

In their analysis Hoadley et. al. report 'extensive multiclonal' seeding in this patient; however, we find that often unidirectional monoclonal seeding perfectly explains the pattern of dissemination and the subclones identified by sciClone project perfectly onto the sample phylogeny. The original claims may stem from the fact that the founder cell(s) of the metastases will contain mutations that have been assigned to separate clusters by the subclonal deconvolution. Such a definition of polyclonal seeding is much too broad, as it would render at least one sample as polyclonally seeded in any study with more than three samples. Instead, we maintain that a metastasis can only be defined as polyclonally seeded when it is founded by cells harbouring mutations from clusters that are not hierarchically related.

In Supplementary Fig. 5B we see that the subclone projection fails as two clusters, 2 and 4, appear unparsimoniously in both the primary and the (apparently) distantly related adrenal and liver metastases. We began our reconciliation of the subclonal deconvolution and the sample tree by noting that cluster 2 appears clonally at higher frequency than cluster 4 in the liver and adrenal metastases. By observation of the sequencing depth ratio at the sites of these mutations, we found evidence that they exist in a region of copy gain that was missed when copy number was determined for these samples. For this reason, we propose that clusters 2 and 4 should be merged.

Closer observation of the primary sample reveals the existence of two subclones, one harbouring mutations in common with the kidney and spinal metastases and one harbouring mutations in common with the spinal and adrenal metastases. Perhaps the most parsimonious explanation for the existence of these two subclones is that two separate metastatic lineages derived from the primary tumour, and that the bulk sample of the primary included subpopulations descended from the founding cells of both of these lineages. This hypothesis would imply either the convergent evolution of a metastatic phenotype, or that the cancer exhibits a 'born to be bad' metastatic subtype with metastatic potential arising very early in the life history of the disease. Such dynamics have recently been reported in a cohort of match primary/metastatic breast cancers[22]. However, without further validation experiments it is not possible to rule out asynchronous seeding from the liver/adrenal metastasis back to the primary site.

Regardless of which of these hypotheses is true, neither implies the existence of extensive polyclonal seeding. The first is entirely concordant with unidirectional, monoclonal seeding, albeit from separate clones within the primary tumour. The second implies only asynchronous metastasis to primary seeding which need not be polyclonal, but, if experimentally verified, would have significant implications for clinical decision regarding resection of the primary tumour.

**Reporting summary.** Further information on research design is available in the Nature Research Reporting Summary linked to this article.

## Data availability

Purity and ploidy data are available from Supplementary Data 5. Copy number, SNV calling, clustering assignment and methylation haplotypes information are available as Supplementary Data 1, 2, 4 and 7 respectively. Sequence data have been deposited at the European Genome-phenome Archive (EGA), which is hosted by the EBI and the CRG, under accession number EGAS00001004014. Further information about EGA can be found on https://ega-archive.org. LEGACY study Research Ethics Committee approval (IRB) number was 13/LO/1535. Breast cancer ITH study (Fig. 4h–k) Research Ethics Committee (IRB) number was 13/LO/1015.

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

## Acknowledgements

The authors acknowledge funding from Breast Cancer Now and the Royal Marsden and ICR NIHR Biomedical Research Centre. A.S. is supported by Wellcome Trust (202778/B/16/Z) and Cancer Research UK (A22909). We also acknowledge funding from the National Institute of Health (NCI U54 CA217376) to A.S. and C.C.M. This work was also supported by a Wellcome Trust award to the Centre for Evolution and Cancer (105104/Z/14/Z). H.T. is supported by Cancer Research UK (A23536). L.Z. is supported by the European Union's Horizon 2020 research and innovation programme under the Marie Skłodowska-Curie Research Fellowship scheme (846614). C.C.M. was supported in part by NIH grants U54 CA217376, P01 CA91955, R01 CA170595, R01 CA185138 and R01 CA140657 as well as CDMRP Breast Cancer Research Program Award BC132057 and an Arizona Investigator Grant ADHS18-198847. We also acknowledge funding of the Pilot LEGACY programme by Breast Cancer Now (Breakthrough Breast Cancer) UK. L.M. is supported by Cancer Research UK (A23110). Image from Figs. 1a and 3a was taken from Servier Medical Art (licensed under a Creative Commons Attribution 3.0 Unported License). A UK patent application has been submitted in relation to the use of methylation clocks (application number 2000747.2).

## Author contributions

G.D.C. and D.N. analysed the data. I.S. generated the data. G.D.C., D.N. and I.S. interpreted the results and contributed to manuscript preparation. H.T., L.Z. and T.H. contributed to data analysis. C.C.M., L.M., G.S. and A.A. helped with the design and supervision of the study. P.B. organised patient recruitment and coordinated tissue sampling. A.S., P.B. and G.S. conceived, designed and supervised the study, and wrote the manuscript.

## Competing interests

G.S. is employee of AstraZeneca UK and also shareholder. A.A. is a co-founder of Tango Therapeutics, Azkarra Therapeutics, Ovibio Corporation; a consultant for SPARC, Bluestar, TopoRx, ProLynx; a member of the SAB of Genentech and GLAdiator; receives grant/research support from SPARC and AstraZeneca; A.A. holds patents on the use of PARP inhibitors held jointly with AstraZeneca which he has benefitted financially (and may do so in the future) through the ICR Rewards to Inventors Scheme.
