## [Peer Review File · Nature Communications]

Reviewers' comments:

Reviewer #1 (Remarks to the Author):

Cresswell et al perform an interesting proof-of-concept study comparing multi-tissue whole genome sequencing profiles with that of ctDNA, and also introduce a mC quantification to ctDNA to facilitate epigenetic profiling in the absence of mutational priors.

Generally I think this study reveals some very promising results and avenues of investigation, but some major concerns remain which point to either serious revisions or consideration of a more specialized journal of case studies.

Concerns:

1) The mutational signature in Fig 1E is very hard to understand, both literally or in its significance to the overall thrust of the paper. It appears to show a near-universal (truncal?) enrichment in an 'aging' signature 1A, which is never defined, nor its concomitant APOBEC signature 2. How do these signatures mesh with Figure 2 C)? Are they over or under-enriched?

2) Referring to the extended commentary starting from the last paragraph of page 4 through to the end of the first paragraph of page 5, it isn't clear why high-quality, but unique, mutations among all metastatic deposits are ignored. Parsimony, monoclonal seeding, is engineered if any site-specific high quality somatic mutations that happened to exist are ignored by construction. 2/12 being polyclonal, including that of the liver met that ultimately killed the patient, is a substantial sign that monoclonal seeding may not be the most important evolutionary modality even if somehow deemed the most common in this patient. Further, it would seem reasonable to ask, given the stated LEGACY rationale, how many metastatic deposits one must sequence in order to obtain a reliable reconstruction of the cascade. Are the 12 used in LEGACY patient 1 really all necessary?

3) Even though only two neoantigens were found in common across all mets, this does not obviously imply there was no preceding or ongoing immune response recruited by met neo-peptide production. The authors must consider using any remaining tumor met DNA to conduct TCR sequencing to definitively confirm the presence/absence of an adaptive immune response which may have either led to a highly edited met profile (hence only 2 neoantigens), or a completely barren profile (because of only 2 antigens). If tissue remains, RNA-seq could also establish considerably more information concerning the adaptive and innate immune state that actually accompanied the claimed modest neoepitope profile. The mC results even state that lymphocytes seem to be confounding the tumor signal!

4) What sites characterize the methylation clocks within ZNF454 and IRX2? Can the methylation profiles be plotted to give the reader a view of the beta profile? Why are these two sites sufficient? How does this patient's ctDNA mC 'drift' compare to what was found earlier? What threshold denotes saturation (beta range)? This is particularly important to discuss more given that LEGACY patient 2 cannot even be analyzed via the methyl clock given this as the stated reason. This lack of feasibility criteria somewhat dampens enthusiasm about the general applicability of the new mC ctDNA technique.

5) Generally the link between clonal evolution and mC is poorly presented, or at least I can't understand it as presented. Graphical presentation suffers as well, e.g. correlation plots in Figure 4G look like they're dominated by a couple of outliers, which would yield a high p value in any spearman correlation. What's the p value?

Reviewer #2 (Remarks to the Author):

In patient 2, how confidence are you in the lineage without the primary tumor to anchor the clone trees? Without a primary tumor, it seems the ability to draw any significant conclusions of the phylogenetic tree challenging.

The VAF cutoff is extremely strict at 10%. I'm assuming that it is because there was no orthogonal validation of the mutation calls in the samples in this study. The "stricter and more conservative bioinformatics analysis" as described in the Figure S3 legend in itself could explain differences between the analyses in this manuscript versus the data from the other manuscript that was reanalyzed. The other paper had clones with VAFs less than 10% in the primary but had validated in orthogonal DNA sequencing as well as validated in RNA sequencing and observed in the metastases. Additionally, the SciClone restriction to a maximum of two clones per tumor would inherently not return the same result as what was described in Hoadley et al which used SciClone with a maximum of 20 clusters allowed.

With a median coverage of 60X-90X, there may be limited ability to find the more rare clones, especially without validation sequencing data. But for the additional clones identified in the mets, did you happen to look and see if they were visible in the primary tumor even though they didn't pass your VAF threshold criteria? The overall purity level was relatively low for many of the samples. The two mets identified with a polyclonal nature were samples with some of the highest purity levels.

For the subclonal reconstruction, the discussion states that only clonal diploid regions were used for the analysis. How much of your data was then used for this analysis? In your samples, many were diploid, but this can be troublesome in more aneuploid samples.

The APOBEC mutational signatures didn't really seem to add useful knowledge to the manuscript.

The methylation data was interesting, but I feel like there could be additional details about the haplotypes you identified and more details about the estimate of the contribution of each met site with ctDNA and methylation data. From the methods, it looks like it was based on the linear model, but did it only include mutations that were specific to each met site or was it all the mutations. There also does seem to be a bias to one met site over others.

The discussion was very weak. There was limited discussion on the ctDNA and methylation which were by far the more interesting parts of this paper. It might also be good to add the thoughts on the polyclonal versus monoclonal seeding especially since there are other data sets that also suggests polyclonal seeding such as Ullah JCI 2018, Escheverria NCOMM 2018, and single cell data Leung Genome Research 2017. I believe that there are many different methods for metastasis and likely one size fits all is not appropriate. It might be important to take some of the language from your methods into your discussion such as your definition of polyclonal seeding

The EGA accession number is missing in the data availability section.

Minor

1. The supplemental figure legends are incredibly space and could use additional information to interpret the figures such as axes labels and colors of points.
2. Figure 4 legend is missing details for E and F.
3. Figure S7 is listed after info about methylation clocks used in breast cancer, but figure and legend says from other healthy tissues from autopsy.

Reviewer #3 (Remarks to the Author):

In this manuscript, the authors introduced a methylation profiling method in ctDNA to deconvolve the cancer cell genealogy at single-molecule resolution without prior knowledge. The use of methylation clocks is innovative and greatly reduces the cost of sequencing while maintaining high resolution, however, some concerns need to be addressed.

Major concerns:

1. Though the proposed methylation clocks worked great in patient 1, and probably would also work in patient 4 and patient 16, they failed in patient 2. The failure percentage seems too high. Could the authors find some other clocks that would fit more patients?
2. When doing methylation clocks analysis, the authors removed samples with prevalence < 1%. Thus, some clones were excluded in Fig 4C-D. But all clones were present in Fig 4E-F again. What prevalence cutoff did the authors use in Fig 4C-D? If 1% is used as cutoff, what would the clone tree be?
3. The authors only used clinical samples with single time point. To better evaluate this approach, it is recommended to apply it to clinical samples with multiple time points.

Minor concerns:

1. The authors should make the manuscript more reader friendly: 1) some acronyms, like CCF and WGD, are not clearly explained in the figure legends or in the context; 2) the meaning of the bars in the clone trees are also not described; and 3) the axis unit in Fig 4G-J is also missing.
2. In the Figure 4 A-D legends, the authors used contribution of "metastatic samples" or "metastatic sites" to ctDNA. Using "tumor samples" is recommended because primary tumors also contributed to ctDNA according to the authors' data.

Reviewer #4 (Remarks to the Author):

This paper describes two case reports where genomic characterisation of several metastatic sites from post mortem tissue has been correlated with cfDNA for both somatic mutations and methylation profiles. The authors suggest that methylation profiling allows tracking evolutionary changes in the cancer cell population at single-molecule resolution without prior knowledge.

Whilst the underlying science is sound the paper was difficult to follow and key information is missing.

Major comments are:

- 1) The study reports on genomic profiles in two post mortem cases of metastatic breast cancer but also includes data from early breast cancers (Figure 4 G-J) and post mortem tissue of healthy donors (Suppl Figure 7). Full information should be provided on the ethical approvals for all patients and samples as a section in the methods.
- 2) The two cases reported are described in the results with a bullet point clinical history. This is unusual - a patient timeline would capture this better and be more informative in relation to the main Figures.
- 3) Orthogonal validation of key mutations would be helpful at least in the tumour tissue.
- 4) More detail is needed for the methylation clock analysis. Authors admit they couldn't do this for

patient 2, so why are they included in the paper?

5) More detail is needed for informatics. Eg. the mutation signature analysis is not described, how many variants were included here? E.g in Figure 1 Mutational signature analysis identifies dominant signature 1A (aging) in all samples for patient 1 does this reflect the number of variants included?

6) Summary figures or heatmaps adding the plasma DNA would be very useful to see how it overlaps with the tissue.

7) The Figures and supplementary figures are difficult to follow from the legends given. More detailed explanations are needed throughout: E.g. Figure 1 is confusing. They list different clusters in 1C, but none of the colours are in the figure? Also, colours for clusters 13, 13 and private need changing as they look the same in the image given. Figure 2 B and C are impossible to interpret from the information given in the legend. What does figure 2C mean? Supp Fig 7 data is given for colon, brain and heart, for autopsy samples from healthy tissues from donors of different ages. No information is given on these individuals - see point 1.

Minor:

The paper needs more thorough proof reading throughout, for example address details missing for one of the senior authors.

Point by point response to reviewers

We are extremely grateful to all reviewers for the positive opinion of our work and their constructive criticism. Below we address each point raised.

Reviewer #1 (Remarks to the Author):

Cresswell et al perform an interesting proof-of-concept study comparing multi-tissue whole genome sequencing profiles with that of ctDNA, and also introduce a mC quantification to ctDNA to facilitate epigenetic profiling in the absence of mutational priors.

Generally I think this study reveals some very promising results and avenues of investigation, but some major concerns remain which point to either serious revisions or consideration of a more specialized journal of case studies.

We thank the reviewer for the positive assessment of our work and we address their concerns below.

Concerns:

1) The mutational signature in Fig 1E is very hard to understand, both literally or in its significance to the overall thrust of the paper. It appears to show a near-universal (truncal?) enrichment in an 'aging' signature 1A, which is never defined, nor its concomitant APOBEC signature 2. How do these signatures mesh with Figure 2 C)? Are they over or under-enriched?

This is a good point and indeed we do recognize that Fig 1E does not add much to the results. It shows that signature 1A, which is the product of deamination at CpG sites due to DNA aging¹, is the dominant signature in this tumour. We have now removed panel E from the figure and just report this finding in the manuscript (lines 130-134, text in red).

2) Referring to the extended commentary starting from the last paragraph of page 4 through to the end of the first paragraph of page 5, it isn't clear why high-quality, but unique, mutations among all metastatic deposits are ignored. Parsimony, monoclonal seeding, is engineered if any site-specific high quality somatic mutations that happened to exist are ignored by construction.

We apologise for the lack of clarity. We did not exclude mutations that are specific to a site, for the clustering analysis with sciClone we just exclude mutations found in only one sample (private in one sample), hence those that have no impact on polyclonality estimates or shared relationships. We then include them in the final analysis when we reconstructed the phylogenetic trees (indeed sample-specific mutations are present in figures 1C and 2A as private mutations are at least informative of age of divergence). We applied the clustering analysis only to mutations found in at least 2 samples because otherwise sciClone tends to group all those mutations into a single cluster, which makes no sense. We have clarified this point in the revised version of the manuscript (see Methods, 'Subclonal reconstruction' section, lines 588-612, text in red).

2/12 being polyclonal, including that of the liver met that ultimately killed the patient, is a substantial sign that monoclonal seeding may be not be the most important evolutionary modality even if somehow deemed the most common in this patient.

This is a very good point, we have now discussed this in the revised manuscript (see Discussion, lines 448-451, text in red).

Further, it would seem reasonable to ask, given the stated LEGACY rationale, how many metastatic deposits one must sequence in order to obtain a reliable reconstruction of the cascade. Are the 12 used in LEGACY patient 1 really all necessary?

This is a very interesting point about power vs number of samples. It is the case that it is possible to reconstruct the metastatic cascade as proposed in Figure 2C by sampling fewer than 12 samples. We propose that that a minimum of 8 samples could reconstruct the cascade between sampled organs. These samples would include a minimum of one sample from each organ (primary breast tissue, lymph, lung, ovarian, diaphragm and liver metastases) however, we propose that the polyclonal liver and ovarian samples are essential to the cascade as well as an additional monoclonal sample from the liver and ovary that would confirm the link between the polyclonal samples and the monoclonal liver/ovary samples.

As the status of polyclonality/monoclonality is unknown prior to sequencing, there is a chance that even if all organs are sampled a minimum of once, eight random samples may not be adequate if key samples are not sequenced (i.e. polyclonal samples). We calculate that of the 128 sampling strategies that would involve taking eight samples and sampling each site at least once, only 12 would gather all required samples to reconstruct the cascade (9%). The chance of achieving the required sample set increases as more samples are taken (9 samples = 25%, 10 samples = 45%, 11 samples = 70%, 12 samples = 100%) – see Figure R1 below.

Figure R1. Proportion of phylogenetic tree solutions that recapitulate the metastatic cascade we observe it (with 12 samples) as function of number of samples taken.

3) Even though only two neoantigens were found in common across all mets, this does not obviously imply there was no preceding or ongoing immune response recruited by met neo-peptide production. The authors must consider using any remaining tumor met DNA to conduct TCR sequencing to definitively confirm the presence/absence of an adaptive immune response which may have either led to a highly edited met profile (hence only 2 neoantigens), or a completely barren profile (because of only 2 antigens). If tissue remains, RNA-seq could also establish considerably more information concerning the adaptive and innate immune state that actually accompanied the claimed modest neopeptide profile.

We thank for the reviewer for this suggestion. However, unfortunately we had very limited leftover DNA from the samples. We did attempt to extract RNA but the degradation was very high as samples were collected post mortem. Although we thank the reviewer for this important point, we maintain that at this point TCR profiling, although interesting, is beyond the scope of this work.

The mC results even state that lymphocytes seem to be confounding the tumor signal!

Sorry for the confusion, there we refer to lymphocytes in the peripheral blood, not in the tissue. We now clarify this in the revised version of the manuscript (line 386, text in red).

4) What sites characterize the methylation clocks within ZNF454 and IRX2? Can the methylation profiles be plotted to give the reader a view of the beta profile? Why are these two sites sufficient? How does this patient's ctDNA mC 'drift' compare to what was found earlier? What threshold denotes saturation (beta range)? This is particularly important to discuss more given that LEGACY patient 2 cannot even be analyzed via the methyl clock given this as the stated reason. This lack of feasibility criteria somewhat dampens enthusiasm about the general applicability of the new mC ctDNA technique.

In new Figure S12, reported as Figure R2 below, we plot the bulk percentage methylation (equivalent of beta values in methylation arrays) for each CpG and each tissue sample. This is the first time we analyse ctDNA with molecular clocks but we found that in this case methylation levels are a bit lower compared to previously analysed colon tissue where cell proliferation is higher and hence methylation clocks are faster².

We use 80% methylation as a cut off for removing saturated haplotypes. This equates to >12 sites methylated in ZNF454 (11 in the short version of the clock) and >7 methylated sites in IRX2. As methylation clocks become fully saturated given infinite time the number of unique haplotypes decreases and therefore the phylogenetic information will be lost. As a large proportion of haplotypes violated this in Patient 2 it was not suitable for analysis.

The results on the mC for patient 2 have been disappointing but frankly it has been the first time we have observed widespread hypermethylation. In all previously analysed cohorts of colorectal cancer this was not frequently observed²⁻⁷.

Figure R2: Percentage methylation of CpG positions in IRX2 (A) and ZNF454 (B) in Patient 1, excluding haplotypes that are both hypo- and hypermethylated. Percentage methylation of CpG positions in IRX2 (C) and ZNF454 (D) in Patient 2, excluding haplotypes that are hypomethylated, saturated haplotypes are retained to display reported saturation.

5) Generally the link between clonal evolution and mC is poorly presented, or at least I can't understand it as presented. Graphical presentation suffers as well, e.g. correlation plots in Figure 4G look like they're dominated by a couple of outliers, which would yield a high p value in any spearman correlation. What's the p value?

We apologise for the suboptimal presentation of mC. We have now extended the explanation of clocks and added a new panel (F) in Figure 4 explaining the concept graphically. We now reported the p-value for the Pearson correlation in the caption of new Figure 4H-K.

Reviewer #2 (Remarks to the Author):

In patient 2, how confidence are you in the lineage without the primary tumor to anchor the clone trees? Without a primary tumor, it seems the ability to draw any significant conclusions of the phylogenetic tree challenging.

Although it would be ideal to have access to the primary tumour sample in patient 2 as well, we note that in the context of metastatic cancers this is generally very difficult because the primary is resected many years earlier. This is why virtually all previously presented samples from autopsy programmes do not analyse the primary tumour⁸⁻¹¹.

It is a special clinical case for Patient 1 where, having been diagnosed with metastatic disease, it was noted that the primary tumour was not a clinically active site and resection was deemed unnecessary. This is incredibly valuable from the point of view of the science but rare as a clinical event.

Having said this, as for the other studies in metastatic cancer, we argue that not having the primary tumour sample does not substantially impact our analysis as reconstructing the most recent common ancestor of all mets still allows to study the metastatic cascade through phylogenetic methods.

We extend the discussion over the relationship between the primary and the metastases in the revised version of the manuscript (lines 440-445, text in red).

The VAF cutoff is extremely strict at 10%. I'm assuming that it is because there was no orthogonal validation of the mutation calls in the samples in this study. The "stricter and more conservative bioinformatics analysis" as described in the Figure S3 legend in itself could explain differences between the analyses in this manuscript versus the data from the other manuscript that was reanalyzed. The other paper had clones with VAFs less than 10% in the primary but had validated in orthogonal DNA sequencing as well as validated in RNA sequencing and observed in the metastases. Additionally, the SciClone restriction to a maximum of two clones per tumor would inherently not return the same result as what was described in Hoadley et al which used SciClone with a maximum of 20 clusters allowed.

We apologise for the lack of clarity. In our analysis we used the number of reads filter and a CCF cutoff, not a VAF cut off. We limited sciClone to 2 times the number of samples minus one (e.g. n=25 clusters for patient 1), which is even more than Hoadley et al (n=20). We now clarified this in the revised version of the manuscript. We did attempt to extract and analyse RNA from those samples, but unfortunately the RNA quality was very low due to collection of tissue post mortem.

We note that we reanalyze the targeted data from Hoadley et al. and found that the conclusions about polyclonal seeding were confounded by missed copy number alterations and assignment of mutational clusters to the sample tree (lines 692-706, text in red).

With a median coverage of 60X-90X, there may be limited ability to find the more rare clones, especially without validation sequencing data. But for the additional clones identified in the mets, did you happen to look and see if they were visible in the primary tumor even though they didn't pass your VAF threshold criteria?

This is a very good point. We checked the variants that were shared amongst the distal metastatic cascade but we found no evidence of those mutations in the primary tumour, as shown in Figure R3 below.

Figure R3. Number of reads with variant for mutations that were found in the distant metastatic samples shows no evidence of those mutations in the primary tumour at current data resolution.

The overall purity level was relatively low for many of the samples. The two mets identified with a polyclonal nature were samples with some of the highest purity levels.

We did check whether we had more power of detecting polyclonality because of purity, but that is not the case, as shown in Figure R4 below and reported in the revised version of the manuscript (lines 187-188, text in red).

Figure R4: Tumour specific coverage, calculated by coverage of the WGS multiplied by the tumour purity shows that polyclonal samples are not limited to the sample with the highest coverage and/or purity.

For the subclonal reconstruction, the discussion states that only clonal diploid regions were used for the analysis. How much of your data was then used for this analysis? In your samples, many were diploid, but this can be troublesome in more aneuploid samples.

As previously argued¹², we leveraged on the hitchhiking principle and whole-genome linkage disequilibrium in cancer and focus on diploid regions only to reduce the noise of CCF estimation from highly aneuploid regions of the genome. This is possible with whole-genome sequencing as there are large amounts of SNVs in diploid regions only. In the case of a highly aneuploid genome, like whole-genome duplications, only triploid or only tetraploid regions can also be considered, as long as the multiple clonal clusters of 1/3 and 2/3 variants are accounted for.

To specifically answer the question, ~65% of the genome was diploid in all samples in Patient 1 and ~59% in Patient 2, showing that as the reviewer pointed out, in these cases a majority of the genome is used for subclonal analysis.

The APOBEC mutational signatures didn't really seem to add useful knowledge to the manuscript.

Yes, we agree with this comment and we have now removed the panel from Figure 1. In Patient 2 two samples contain mutations in DNA damage repair proteins and subsequently we observed alterations in their APOBEC signatures, this result may be of interest to researchers interested in the biochemistry of APOBEC signature repair, however we agree that this result is not central to the manuscript.

The methylation data was interesting, but I feel like there could be additional details about the haplotypes you identified and more details about the estimate of the contribution of each met site with ctDNA and methylation data. From the methods, it looks like it was based on the linear model, but did it only include mutations that were specific to each met site or was it all the mutations.

We apologise for the lack of clarity. Both in the methylation analysis and in the SNV contribution to ctDNA we used all the data from all the samples. We now clarified this in the revised version of the manuscript (line 298, text in red). It is only for the subclonal analysis of SNVs we did first exclude mutations that are present in a single sample. This is to avoid sciClone to output too many false clusters. We add those at the end of the analysis anyway in Figure 2A (sample-specific tree branches).

There also does seem to be a bias to one met site over others.

Yes indeed one of our striking findings is that the methylation data on the contribution of DNA from different metastatic sites largely recapitulates what we find orthogonally with the SNVs from whole-genome analysis, with the liver in patient 1 and multiple lymph nodes in patient 2 mainly contributing to ctDNA.

The discussion was very weak. There was limited discussion on the ctDNA and methylation which were by far the more interesting parts of this paper. It might also be good to add the thoughts on the polyclonal versus monoclonal seeding especially since there are other data sets that also suggests polyclonal seeding such as Ullah JCI 2018, Escheverria NCOMM 2018, and single cell data Leung Genome Research 2017. I believe that there are many different methods for metastasis and likely one size fits all is not appropriate. It might be important to take some of the language from your methods into your discussion such as your definition of polyclonal seeding

Excellent point. We have now extended the discussion and included the above references (lines 451-454, text in red).

The EGA accession number is missing in the data availability section.

Apologies, all the data are now been uploaded online under EGA accession number EGAS00001004014. Upload should be done in the next few days.

Minor

1. The supplemental figure legends are incredibly space and could use additional information to interpret the figures such as axes labels and colors of points.

OK done.

2. Figure 4 legend is missing details for E and F.

OK done.

3. Figure S7 is listed after info about methylation clocks used in breast cancer, but figure and legend says from other healthy tissues from autopsy.

Yes indeed we used autopsy data from healthy tissues to show the clock-like behavior of these methylation sites. A good methylation clock should accumulate CpG methylation in dividing tissue like the colon, but remain at low methylation in non-dividing tissues. We clarified this aspect further in the revised version of the manuscript (lines 375-378, text in red).

Reviewer #3 (Remarks to the Author):

In this manuscript, the authors introduced a methylation profiling method in ctDNA to deconvolve the cancer cell genealogy at single-molecule resolution without prior knowledge. The use of methylation clocks is innovative and greatly reduces the cost of sequencing while maintaining high resolution, however, some concerns need to be addressed.

We thank the reviewer for appreciating the novelty and advancements we present in the paper.

Major concerns:

1. Though the proposed methylation clocks worked great in patient 1, and probably would also work in patient 4 and patient 16, they failed in patient 2. The failure percentage seems too high. Could the authors find some other clocks that would fit more patients?

Indeed, it was a surprise that the clocks we extensively used in the past did not work for Patient 2. Widespread hypermethylation was not found in previous cohorts of colorectal and brain cancer^{2,13}. We are now in the process of testing new molecular clocks that may be promising but for now the validation phase is still ongoing.

2. When doing methylation clocks analysis, the authors removed samples with prevalence < 1%. Thus, some clones were excluded in Fig 4C-D. But all clones were present in Fig 4E-F again. What prevalence cutoff did the authors use in Fig 4C-D? If 1% is used as cutoff, what would the clone tree be?

Apologies for the misunderstanding. We used the 1% cut off for the phylogenetic analysis in panels E and F as we wished to use the most abundant haplotypes to construct the tree and minimize homoplasmy, however for the linear modelling results in C and D we did not use the filter as we believed the linear modelling would be more robust to very rare haplotypes and if we artificially removed these haplotypes we may have biased the fit. We now clarified this in the revised version of the manuscript (lines 298-300, text in red and 'Methylation clock analysis' section).

3. The authors only used clinical samples with single time point. To better evaluate this approach, it is recommended to apply it to clinical samples with multiple time points.

It would be fantastic to have multiple time points and indeed we are now extending our analysis of other cohorts of live patients where we can collect longitudinal data. However, in this context of a rapid autopsy unfortunately we have no more chances of collecting additional samples.

Minor concerns:

1. The authors should make the manuscript more reader friendly: 1) some acronyms, like CCF and WGD, are not clearly explained in the figure legends or in the context; 2) the meaning of the bars in the clone trees are also not described; and 3) the axis unit in Fig 4G-J is also missing.

Good point, we have now addressed this in the revised version of the manuscript and figures.

2. In the Figure 4 A-D legends, the authors used contribution of “metastatic samples” or “metastatic sites” to ctDNA. Using “tumor samples” is recommended because primary tumors also contributed to ctDNA according to the authors’ data.

Good point, corrected.

Reviewer #4 (Remarks to the Author):

This paper describes two case reports where genomic characterisation of several metastatic sites from post mortem tissue has been correlated with cfDNA for both somatic mutations and methylation profiles. The authors suggest that methylation profiling allows tracking evolutionary changes in the cancer cell population at single-molecule resolution without prior knowledge.

Whilst the underlying science is sound the paper was difficult to follow and key information is missing.

We thank for appreciating the science and apologies for the difficulty in following the manuscript. We have now revised the paper to make the reading easier, as well as addressed the points below.

Major comments are:

1) The study reports on genomic profiles in two post mortem cases of metastatic breast cancer but also includes data from early breast cancers (Figure 4 G-J) and post mortem tissue of healthy donors (Suppl Figure 7). Full information should be provided on the ethical approvals for all patients and samples as a section in the methods.

Agreed, apologies for the missing information. We have now explicitly added all the information in the revised version of the manuscript under the Data Access section.

2) The two cases reported are described in the results with a bullet point clinical history. This is unusual - a patient timeline would capture this better and be more informative in relation to the main Figures.

Good point, we have now done it in new Figure S1.

3) Orthogonal validation of key mutations would be helpful at least in the tumour tissue.

Given the number of variants we can detect with whole-genome sequencing, the depth of sequencing, as well as the robustness of the phylogenetic analysis indicated by high bootstrap values for the trees (Figure 2A and 3E), we argue that it is very unlikely that the analysis is significantly influenced by false positive variants. However, to address the reviewer comment, we have tested driver mutation PIK3CA E542K in patient 1 using ddPCR and compared with VAF values inferred with WGS, confirming the results. As shown in Figure R6 below, ddPCR validated

the variant found in WGS, although underestimated its VAF probably due to bias in the affinity of variant probe in ddPCR. Indeed, we verified that the VAF estimated from WGS was the correct one by comparing PIK3CA VAF from WGS versus purity independently estimated with the copy number profiles (Figure R7 below). We now report this in the revised version of the manuscript and added the results as new Table S3 (lines 105-106, text in red).

Figure R6: Variant allele frequency of the PIK3CA E542K driver mutation derived from whole genome sequencing in Patient 1 with binomial confidence intervals (Wilson method, x-axis) compared to digital droplet polymerase chain reaction (ddPCR) fractional abundance with Poisson fractional abundance confidence intervals (y-axis) as provided by Quantasoft™.

Figure R7: Variant allele frequency (VAF) of the PIK3CA E542K driver mutation derived from whole genome sequencing in Patient 1 with binomial confidence intervals (Wilson method, y-axis) compared to purity estimates derived from copy number analysis. As the PIK3CA mutation is in two copies of all cancer cells, the VAF is a direct estimate of tumour purity. This is independently validated by copy number analysis purity estimates.

4) More detail is needed for the methylation clock analysis. Authors admit they couldn't do this for patient 2, so why are they included in the paper?

We apologise for the lack of clarity. Indeed, we could not use the methylation data for patient 2 as they were unexpectedly hypermethylated. We do not analyse the methylation data from this patient in the manuscript, however to address a comment by Rev #1 we just report the beta values for both patients in new Figure S12, showing more clearly the hypermethylation problem for patient 2. We also added more details about the methylation analysis in the revised version of the manuscript.

5) More detail is needed for informatics. Eg. the mutation signature analysis is not described, how many variants were included here? E.g. in Figure 1 Mutational signature analysis identifies dominant signature 1A (aging) in all samples for patient 1 does this reflect the number of variants included?

Apologies, we have now addressed this in the revised version of the manuscript (lines 566-570, text in red).

6) Summary figures or heatmaps adding the plasma DNA would be very useful to see how it overlaps with the tissue.

Very good point, we have now added these analyses as Figure S9 and refer it in the text.

7) The Figures and supplementary figures are difficult to follow from the legends given. More detailed explanations are needed throughout: E.g. Figure 1 is confusing. They list different clusters in 1C, but none of the colours are in the figure? Also, colours for clusters 13, 13 and private need changing as they look the same in the image given. Figure 2 B and C are impossible to interpret from the information given in the legend. What does figure 2C mean? Supp Fig 7 data is given for colon, brain and heart, for autopsy samples from healthy tissues from donors of different ages. No information is given on these individuals - see point 1.

We apologise for the lack of clarity. We have now changed Figure 1C to clearly reflect the mutational clusters. We also modified the legend of Figure 1C and 2B,C (text in red). We also describe better Figure S7 (now Figure S10) and clarify the results (lines 779-782, text in red).

Minor:

The paper needs more thorough proof reading throughout, for example address details missing for one of the senior authors.

Apologies, we have now proofread the paper and hopefully removed all typos and fix the senior author's missing information.

References

1. Alexandrov, L. B. *et al.* Clock-like mutational processes in human somatic cells. *Nature Genetics* **47**, 1402–1407 (2015).
2. Sottoriva, A., Spiteri, I., Shibata, D., Curtis, C. & Tavaré, S. Single-molecule genomic data delineate patient-specific tumor profiles and cancer stem cell organization. *Cancer Res.* **73**, 41–49 (2013).
3. Siegmund, K. D., Marjoram, P., Tavaré, S. & Shibata, D. Many colorectal cancers are 'flat' clonal expansions. *Cell Cycle* **8**, 2187–2193 (2009).
4. Yatabe, Y., Tavaré, S. & Shibata, D. Investigating stem cells in human colon by using methylation patterns. *Proc. Natl. Acad. Sci. U.S.A.* **98**, 10839–10844 (2001).
5. Nicolas, P., Kim, K.-M., Shibata, D. & Tavaré, S. The stem cell population of the human colon crypt: analysis via methylation patterns. *PLoS Comput. Biol.* **3**, e28 (2007).

6. Graham, T. A. *et al.* Use of methylation patterns to determine expansion of stem cell clones in human colon tissue. *Gastroenterology* **140**, 1241–1250.e1–9 (2011).
7. Humphries, A. *et al.* Lineage tracing reveals multipotent stem cells maintain human adenomas and the pattern of clonal expansion in tumor evolution. *PNAS* **110**, 2490–2499 (2013).
8. Hoadley, K. A. *et al.* Tumor Evolution in Two Patients with Basal-like Breast Cancer: A Retrospective Genomics Study of Multiple Metastases. *PLoS Med* **13**, e1002174 (2016).
9. Savas, P. *et al.* The Subclonal Architecture of Metastatic Breast Cancer: Results from a Prospective Community-Based Rapid Autopsy Program "CASCADE". *PLoS Med* **13**, e1002204 (2016).
10. Gudem, G. *et al.* The evolutionary history of lethal metastatic prostate cancer. *Nature* (2015). doi:10.1038/nature14347
11. De Mattos-Arruda, L. *et al.* The Genomic and Immune Landscapes of Lethal Metastatic Breast Cancer. *Cell Rep* **27**, 2690–2708.e10 (2019).
12. Williams, M. J. *et al.* Quantification of subclonal selection in cancer from bulk sequencing data. *Nature Genetics* **50**, 895–903 (2018).
13. Piccirillo, S. G. M. *et al.* Contributions to drug resistance in glioblastoma derived from malignant cells in the sub-ependymal zone. *Cancer Res.* **75**, 194–202 (2015).

REVIEWERS' COMMENTS:

Reviewer #1 (Remarks to the Author):

I am satisfied with the revisions the authors have provided: the majority of my major concerns have been appropriately addressed. I think it's an interesting contribution and congratulate the authors on some nice work.

Reviewer #2 (Remarks to the Author):

Thank you for addressing my comments.

I think it might be good to add to the text your comment of using ~65% of the genome in patient 1 and ~59% of the genome in patient 2 for the subclonal reconstruction.

While the discussion was added to, nothing was added about the ctDNA and methylation.

Reviewer #3 (Remarks to the Author):

The authors have addressed all the comments raised previously; no additional comments.

Reviewer #4 (Remarks to the Author):

The manuscript is substantially improved by the revisions to date. However there are outstanding issues to address as follows:

Line 104: This variant validated in all samples using 105 digital droplet PCR (Table S3). Needs the word "was" adding between variant and validated

Line 206-220. Figure 2 legend is now much more comprehensive and logical thank you. However, 2B still requires further explanation as to how the different clusters are identified as monoclonal or polyclonal- the colours used are also difficult to distinguish particularly for clusters 6 and 8.

Lines 296-297. Explain what is meant by ctDNA purity here? - How did you estimate this? was it related to mean tVAF of truncal driver mutations ?

Lines 323 to 327 Figure 4. Please present the data described here in 4G onwards as this is very obscure and difficult to follow. "The same methylation clock analysis has been applied to an orthogonal cohort of early breast cancer patients with matched primary, lymph node and ctDNA. High correlation (Pearson) between frequency of methylation haplotypes in the primary tumour of patient 6 was found for both clocks ZNF454 ($p=4.94 \times 10^{-63}$)

Discussion, there is still no consideration of patient 2, please update this.

Point by point response to reviewers

Again, we are extremely grateful to all reviewers for the positive opinion and constructive criticism. Below we address the final points raised by reviewers #2 and #4.

Reviewer #2 (Remarks to the Author):

I think it might be good to add to the text your comment of using ~65% of the genome in patient 1 and ~59% of the genome in patient 2 for the subclonal reconstruction.

Yes, important point. We have now report this in the revised version of the manuscript (page 3, text in red)

While the discussion was added to, nothing was added about the ctDNA and methylation.

Apologies for this, we have added additional discussion on ctDNA and methylation (page 8, text in red).

Reviewer #4 (Remarks to the Author):

The manuscript is substantially improved by the revisions to date. However there are outstanding issues to address as follows:

We thank the reviewer once again for the positive assessment of our work and we address their concerns below.

Line 104: This variant validated in all samples using 105 digital droplet PCR (Table S3). Needs the word "was "adding between variant and validated

Apologies, error corrected.

Line 206-220. Figure 2 legend is now much more comprehensive and logical thank you. However, 2B still requires further explanation as to how the different clusters are identified as monoclonal or polyclonal- the colours used are also difficult to distinguish particularly for clusters 6 and 8.

Sorry for the lack of clarity regarding the colours. We have now changed those and greyed out the clones that are not relevant for the panel (see new Figure 2B).

Lines 296-297. Explain what is meant by ctDNA purity here? - How did you estimate this? was it related to mean tVAF of truncal driver mutations ?

Sorry for the lack of clarity. The purity for the ctDNA samples was determined by solving the copy number profiles of both samples (see Materials and Methods, page 10).

Lines 323 to 327 Figure 4. Please present the data described here in 4G onwards as this is very obscure and difficult to follow. "The same methylation clock analysis has been applied to an orthogonal cohort of early breast cancer patients with matched primary, lymph node and ctDNA. High correlation (Pearson) between frequency of methylation haplotypes in the primary tumour of patient 6 was found for both clocks ZNF454 ($p=4.94 \times 10^{-63}$)

We apologise for the lack of clarity. We have now extended the description of the figure panels in the revised version of the manuscript (page 7, text in red).

Discussion, there is still no consideration of patient 2, please update this.

We have now extended the discussion to consider the results from patient 2 (page 8, text in red).